# Enhancement of Textural and Sensory Characteristics of Wheat Bread Using a Chickpea Sourdough Fermented with a Selected Autochthonous Microorganism

**DOI:** 10.3390/foods12163112

**Published:** 2023-08-18

**Authors:** Chrysanthi Nouska, Magdalini Hatzikamari, Anthia Matsakidou, Costas G. Biliaderis, Athina Lazaridou

**Affiliations:** Department of Food Science and Technology, School of Agriculture, Aristotle University of Thessaloniki, P.O. Box 235, 54124 Thessaloniki, Greece; magdah@agro.auth.gr (M.H.); matsakidou@chem.auth.gr (A.M.); biliader@agro.auth.gr (C.G.B.); athlazar@agro.auth.gr (A.L.)

**Keywords:** fermented chickpeas, sourdough bread, bread texture, starch retrogradation, bread shelf life, sourdough bread organic acids, FTIR protein secondary structure, sensory evaluation

## Abstract

A traditional Greek sourdough, based on the fermentation of chickpea flour by an autochthonous culture, was evaluated as a wheat bread improver. The dominant indigenous microflora (*Clostridium perfringens* isolates) was identified by 16S rDNA analysis, and a selected strain (*C. perfringens* CP8) was employed to ferment chickpea flour to obtain a standardized starter culture (sourdough) for breadmaking. In accordance with toxin-typed strain identification, all isolates lacked the cpe gene; thus, there is no concern for a health hazard. Loaf-specific volumes increased with the addition of liquid, freeze-dried, and freeze-dried/maltodextrin sourdoughs compared to control bread leavened by baker’s yeast only. Following storage (4 days/25 °C), the amylopectin retrogradation and crumb hardness changes (texture profile analysis) revealed a lower degree of staling for the sourdough-fortified breads. Modifications in the protein secondary structure of fortified doughs and breads were revealed by FTIR analysis. High amounts of organic acids were also found in the sourdough-supplemented breads; butyric and isobutyric acids seemed to be responsible for the characteristic ‘butter-like’ flavor of these products (sensory analysis). Overall, the addition of liquid or freeze-dried chickpea sourdough in wheat bread formulations can improve the specific volume, textural characteristics, and sensorial properties of loaves, along with extending bread shelf life.

## 1. Introduction

Sourdough is a mixture of water and flour, often fermented by the autochthonous microflora of the flour [1,2,3], and is one of the oldest methods of cereal fermentation for bread and other baked products in order to improve their shelf life as well as their structural and sensory characteristics. Sourdough fermentation may also affect the nutritional properties of bread by reducing the glycemic index, increasing the uptake of vitamins, minerals, etc., and improving the physicochemical and functional properties of the dietary fiber components [4]. Additionally, fermentation can provide extra nutritionally active compounds such as γ-amino butyric acid and bioactive peptides [5,6]. Bread’s shelf life is relatively short due to physicochemical changes (resulting in firming and staling as well as loss of volatiles) and spoilage caused mostly by molds [7,8,9] and bacteria such as bacilli [10]. The extension of shelf life is thus of great importance for the industry and can be achieved by the addition of chemical preservatives and improvers, or alternatively, by the incorporation of sourdough in the wheat flour dough formulation. It is well documented that sourdough inhibits microbial spoilage due to its acidification and the presence of several antimicrobial compounds [11,12], such as organic acids and bacteriocins, while it can delay staling events due to the production of exopolysaccharides and enzymatic activities related to the hydrolysis of starch and the interruption of the continuous gluten protein matrix [13,14].

Legumes such as chickpeas, peas, and beans have high nutritional quality (valuable protein and starch sources), and they are widely consumed in different forms as food staples worldwide. It is postulated that legumes can be an important source of dietary proteins that range from 20 to 40% of kernel dry weight. Consumption of legumes can be an alternative to animal protein, when the latter cannot be consumed for economic, religious, cultural, or ethical reasons [15]. It is also well known that cultivation of grain legumes has a much lower environmental impact than animal protein production, both per kg of product and per kg of protein produced; the differences become even more pronounced when comparing the climate impact on grain legume cultivation with animal husbandry and meat production (e.g., from beef), as well as taking into account other significant differences for environmental indicators related to eutrophication of aquatic ecosystems, acidification, consumption of energy from fossil resources (oil, natural gas, etc.), and the much greater usage of water and agricultural land. Additionally, considering the growing number of consumers who are either diagnosed with celiac disease or sensitive to consumption of some proteins from cereal grains (e.g., wheat gluten, barley and rye prolamins, etc.), thus necessitating the production of gluten-free bakery products, the use of sourdough in non-wheat baked formulations can often improve their nutritional quality and product shelf life when mixed with other gluten-free ingredients [16].

In Mediterranean countries, fermented chickpeas have been traditionally used as a leavening agent for bread and rusks [17,18]. Specifically in Greece, fermented chickpeas have been used traditionally to produce a type of leavened wheat bread called “eftazymo”, a name derived from the Greek word “aftozymo”, which means fermented by itself [17]. In our previous studies, a gluten-free sourdough made by an autochthonous starter from fermented chickpea extract was found to be an effective leavening and anti-staling agent for improving the quality and extending the shelf life of gluten-free breads [19,20]. In fermented chickpea sourdoughs, the dominant autochthonous microflora has been identified to consist mostly of populations of bacilli and clostridia [21], with *Clostridium perfringens* being responsible for gas production, which is necessary for leavening, and the production of butyric acid that contributes to its characteristic flavor [18,20]. A weakness of the traditional process of chickpea sourdough production, which is not always successful, often arises from the difficulty of preserving the sourdough preparation in its liquid form, hence the need to use it immediately for breadmaking. Moreover, back slopping, a common practice followed in the production of wheat sourdoughs (lactic acid fermentation) that keeps them active for a long period, would enhance the lactic acid bacteria’s dominance over the clostridial microflora, thus changing the characteristic sensory properties of the end-product.

The aim of the present study was to standardize chickpea sourdough production using selected isolated strains from the autochthonous microflora of fermented chickpeas and their employment in the preparation of starter cultures to ferment chickpea flour. To the best of our knowledge, the production of chickpea sourdough wheat bread with a starter culture isolated from the autochthonous microflora of chickpea sourdough has not yet been reported. Isolated strains of clostridia, as identified by their 16S rDNA sequence, were further classified into different toxinotypes and checked especially for the *cpe* gene that expresses an enterotoxin (CPE), causing food poisoning when vegetative cells of *C. perfringens* reach the human gastrointestinal tract and sporulate therein. Then, liquid (fresh) and freeze-dried (with or without addition of maltodextrin) sourdoughs made by a selected *C. perfringens* strain as a starter were evaluated for their leavening, anti-staling, and antifungal potential in wheat breadmaking. More specifically, the examined quality attributes of the sourdough-fortified wheat breads were loaf volume, moisture content, crumb and crust texture, color, and organic acid profile. Additionally, sensory analysis and evaluation of shelf life, concerning mold growth, were conducted. The development and evolution of the dough (before and after proofing) and bread (freshly prepared and after storage) protein network secondary structure was also monitored to gain some insight on the impact of different sourdough types on protein conformational changes and their potential influence on the measured quality attributes.

## 2. Materials and Methods

### 2.1. Raw Materials

Wheat flour (type 55; 11% protein and 0.70% ash content) was gifted by Flourmills Thrakis I. and Ouzounopoulos, SA (Alexandroupoli, Greece), whereas chickpea seeds were supplied from a local farmer (Giannitsa, Greece).

### 2.2. Preparation of Traditional Chickpea Sourdough

Chickpea sourdough was prepared as described by Hatzikamari et al. [17] with slight modifications. Chickpea seeds were milled into flour (Faribon 600 professional, Omas, San Giorgio delle Pertiche, Padova, Italy), instead of being coarsely ground (traditional handling), mixed with boiled tap water (proportion 1:6 *w*/*w*), and incubated at 37 °C for 18 h to generate a fermented dough material (autochthonous microflora). Subsequently, more wheat flour was added to form a batter (3 parts sourdough to 1 part wheat flour) and fermented for another hour at 37 °C until it doubled in volume. All experiments were conducted in triplicate (Figure 1).

### 2.3. Isolation and Identification of Indigenous Clostridial Strains

Portions of 10 g of each sourdough sample were homogenized with 90 mL of sterile peptone diluent (0.1% *w*/*v* peptone, BIOKAR Diagnostics, Allonne, Oise, France). Serial decimal dilutions were inoculated on SPS (Sulfite Polymyxin Sulfadizine) agar (Condalab, Madrid, Spain) and incubated at 37 °C for 5 days under anaerobic conditions (Gas-Pak system, Merck, Darmstadt, Germany). Ten colonies were then transferred to thioglycolate broth, incubated overnight at 37 °C, and purified by streak plating on SPS agar plates that were incubated anaerobically. Single black colonies of clostridia cultured in 10 mL of fresh thioglycolate broth were finally stored in prepared cooked meat medium (BIOKAR Diagnostics, Allonne, Oise, France) following sealing with 1 mL of sterile liquid paraffin to prevent oxygen penetration and kept at room temperature.

The isolated strains were analyzed with SDS-PAGE of their cellular protein fraction as well as by sequencing of the 16S rDNA amplicons. They were also examined for the toxin encoding gene and further evaluated for their gas production and enhanced flavor notes.

#### 2.3.1. Identification of Clostridia Isolates by SDS-PAGE of Cellular Proteins and 16S rDNA PCR

Whole cell proteins of the isolated strains along with a reference strain (*C. perfringens* DSM 756) were subjected to SDS-PAGE, as described elsewhere [22], and their classification was performed by comparison of their protein patterns to the fingerprints of the reference strain.

16S rDNA PCR was conducted in three (CP1, CP8, CP3) out of ten isolates with the primer set of 27F (5′-AGA GTT TGA TCC TGG CTC AG-3′) and 1492R (5′-GGY TAC CTT GTT ACG ACT T-3′), and the amplified products were sequenced (QACS Lab, Metamorfosi, Greece). Homology searches of the 16S rDNA sequences were performed using the GenBank database and the BLASTN algorithm (http://www.ncbi.nlm.nih.gov/-BLAST/ accessed on 13 May 2022).

#### 2.3.2. Toxinotyping Test by Multiplex PCR

Ten strain isolates (CP1-CP10) and *Clostridium perfringens* DSM 756 (German Culture Collection, Brawnschweig, Germany) were activated with two consecutive subcultures in 10 mL TGY (Tryptone Glucose Yeast) broth (tryptone 30 g/L, glucose 5 g/L, yeast extract 20 g/L, cysteine-HCl 0.5 g/L) and incubated overnight at 37 °C. After incubation, the cells were harvested by centrifugation for 5 min at 12,000× *g* (Eppendorf, Microcentrifuge 5415, Wien, Austria) and washed twice with saline solution. To prepare the DNA template, the pellet was suspended in 200 μL of sterile distilled water, incubated at 95 °C for 20 min, centrifuged for 3 min at 12,000× *g*, and ice-cooled. The supernatant was transferred to a sterilized Eppendorf tube and kept in the freezer until analysis [23].

The molecular toxinotyping test was performed by multiplex RCR, according to Ahsani et al. [24]. Four primer sets (Table 1) specific for toxin α, β, ε, and CPE were used to clarify if the isolated strains carried the corresponding structural gene for toxin production [25]. The CPE toxin producing gene was also tested with a second primer set, producing a different size of PCR product [26] (Table 1). The PCR reactions were performed in a 50 μL total volume containing 5 μL of 10× PCR buffer (10 mM Tris-HCL, pH 9.0, 50 mM KCl), 2 μL of 50 mM MgCl_2_, 250 μM of each deoxynucleotide triphosphate, 5 U of Taq DNA polymerase, 100 pmol of each primer, and 3 μL of template DNA [24]. The PCR was conducted in a Thermal Cycler (Biorad laboratories, Hercules, CA, USA) with 35 cycles under the following conditions: denaturation at 94 °C for 45 s, annealing at 55 °C for 30 s, amplification at 72 °C for 90 s, and final extension at 72 °C for 10 min. Finally, the PCR products were electrophoresed on a 1.5% agarose gel in TAE (Tris base, acetic acid, and EDTA) buffer (10×). Visualization and capture of gel images have been made under UV light (Minibis, DNR Bio-Imaging Systems, Jerusalem, Israel).

### 2.4. Fermentation of Chickpea Flour with Clostridium Isolates as Starter Cultures

The selected strain *C. perfringens* CP8, activated with two consecutive subcultures in 10 mL TGY, was incubated for 18 h at 37 °C, and the cells were harvested by centrifugation at 5000× *g* for 5 min and suspended in sterile peptone 0.1% *w*/*v*. Chickpea sourdough was prepared by mixing chickpea flour with boiled tap water (proportion 1:6 *w*/*w*). The mixture was left to cool to 40 °C and then inoculated with *C. perfringens* CP8 at a concentration of ca. 6 log cfu/g and fermented at 37 °C for 2 h. Subsequently, wheat flour was added to form a batter (3 parts sourdough to 1 part wheat flour, *w*/*w*) and fermented for another hour at 37 °C until it doubled in volume; this liquid-like sourdough is abbreviated hereafter as LS. All experiments were conducted in triplicate (Figure 1).

### 2.5. Dried Sourdough Making

For drying, more wheat flour was added to LS (proportion 1:1 *w*/*w*) to form a tight dough, and drying took place in an oven at 35 °C for 48 h under vacuum (Karl Kolb, Scientific technical supplies D-6072, Dreieich, Germany) or freeze-dried (SCANVAC, Coolsafe 100–9 Pro, Labogene, Lynge, Denmark) for 48 h. Moreover, LS was also freeze-dried, as it was in its liquid form, without the addition of extra flour. In all processes, some low-molecular-weight carbohydrates were employed, such as maltodextrin (Tate & Lyle, DE: 17–19), trehalose (Merck, Darmstadt, Germany), and sucrose (Merck, Darmstadt, Germany), separately or in combinations (Appendix A), to act as protective agents for cells during dehydration (drying aids). The viability of clostridia was determined after drying with the plate count technique in SPS agar. The dried sourdough with the highest viability was ground in a mill and stored under anaerobic conditions in sealed polyethylene bags (30 cm × 20 cm) at room temperature. Cell viability measurements were carried out every month for a total of 5 months.

### 2.6. Total Titratable Acidity and pH Determination

For Total Titratable Acidity (TTA), 10 g of dried LS (LSD) and 90 mL of distilled water were homogenized for 3 min in a stomacher bag, titrated with 0.1 M NaOH until reaching a pH of 8.5 [27], and expressed as the required amount (mL) of 0.1 M NaOH (Merck, Darmstadt, Germany). The pH values were determined by direct immersion of the glass electrode of a pH meter (Crison GLP21, Barcelona, Spain) in the LS homogenates.

### 2.7. Breadmaking Procedure

Wheat bread loaves were made from 280 g of dough fortified with liquid (LS) and freeze-dried sourdoughs (without the addition of extra wheat flour prior to freeze-drying), without (FDS) or with 15% (on an LS basis) maltodextrin (FDSM); the latter formulation was examined as the addition of maltodextrin offered the highest viability for *C*. *perfringens* CP8 cells after the freeze-drying process. Wheat bread with baker’s yeast was also prepared to serve as a control. The proportions of components present in all bread formulations are presented in Table 2.

The doughs were made by mixing all the ingredients (Resto Italia SK 10 MO, Urbino, Italy) for 45 min and leaving them for proofing at 40 °C and a relative humidity of 75% for 30 min for control breads and 2 h for sourdough breads to achieve a doubling of the dough volume. Before and after proofing, counts of *C. perfringens* CP8 were determined on SPS agar. Subsequently, breads were baked in an oven (Electrolux, air-o-steam touchline, Stockholm, Sweden) for 25 min and cooled for 2 h before any analysis. Results are presented as means ± SD of three independent experiments.

### 2.8. Physical Properties of Breads

Changes in the texture of breads are related to staling events. Crumb and crust texture, during storage at 25 °C, at three different time intervals (0, 1, and 4 days) for control bread and at four different time intervals (0, 1, 4, and 6) for sourdough breads, were examined by a Texture Analyzer TA.XT plus (Stable Micro Systems, Godalming, Surrey, UK) using the software Exponent Connect Ink Version 8. For control breads, there was no measurement on day 6 since mold growth on their surfaces has been noted.

To evaluate the texture of the crumb, a circular cutter was used to form a cylindrical bread crumb specimen (40 mm diameter × 30 mm height). These samples were subjected to TPA using a 75 mm diameter compression platen probe (Stable Micro Systems, Godalming, Surrey, UK), at 60% applied deformation, 0.8 mm/s test speed, and a delay time of 5 s between the two compression cycles. The textural parameters were estimated according to Armero et al. [28]. Crust texture was evaluated by a penetration test; pieces of the upper crust of breads (40 mm length × 30 mm width × 5 mm thick), were subjected to penetration using a 6.35 mm spherical stainless probe (Stable Micro Systems, Godalming, UK) at a speed of 1 mm/s. The peak force of the recorded force-deformation curve indicates the hardness of the crust.

The loaf volume was determined with a Volscan profiler VSP600 (Stable Micro Systems, Godalming, UK), and the bread-specific volume was calculated in mL/g.

The moisture content of bread crust and crumb samples was evaluated according to the AACC 44-15.02 method [29]. Briefly, approximately 2 g of crumb and crust (separated from the crumb) were weighted and oven-dried at 130 °C for 2 h. The moisture content of crumb and crust was evaluated in fresh specimens (2 h) and after 1, 4, and 6 days (only for sourdough breads) of storage at 25 °C as follows:$$Moisture\;content\;(\%)=\frac{W1-W2}{W1},$$

W1: Weight of crust or crumb before drying;

W2: Weight of crust or crumb after drying.

Color parameters of the crust, according to the CIELAB system (L*, a*, b*), were measured on 5 different spots of the bread loaves by a chromameter (Konica Minolta, CR-400 series, Tokyo, Japan). Calibration was conducted with a white tile (L* = 96.9, a* = −0.04, b* = 1.84).

### 2.9. Organic Acids Analysis by HPLC

#### 2.9.1. Sample Preparation

Bread crumb and sourdough sample preparation for the HPLC was conducted by adding 2 g of the sample to 4 mL of 0.00525 N H_2_SO_4_ aqueous solution, vortexed for 30 s, sonicated by using an ultrasonicator for 15 min and centrifugated at 14,000× *g* for 10 min at 4 °C. The supernatant was filtered through a 0.22 μm millipore membrane filter (25 mm syringe filter, BGB analytik, Boeckten, Switzerland), and the filtrate was used for HPLC analysis.

#### 2.9.2. Equipment and Analytical Procedure

The quantification of organic acids was conducted by High-Performance Liquid Chromatography with a UV-Visible Diode Array detector (UV 6000 LP detector-Spectra SYSTEM, Finningan Mat, Thermo Separation Products, San Jose, CA, USA). Separation was carried out with a Repromer H column (300 × 8 mm) (Dr. Maisch, Ammerbuch, Germany). The mobile phase was 0.0052 N H_2_SO_4_ (sulphuric acid 95–97%, Merck, Darmstadt, Germany) aqueous solution, previously degassed with Helium (99.996% purity, AERIALCO HELLAS, Thessaloniki, Greece), and pumped with a flow rate of 0.2 mL/min through a Marathon IV HPLC Pump (Spark Holland, Emmen, The Netherlands). The temperature of the column (50 °C) was held constant by an insulated column oven (Timberline, TL-50 Controller, Granger, IN, USA). Aliquots of 20 μL of sample were injected in the column, and the detection of organic acids was monitored at 215 nm; their quantification was elaborated via the ChromQuest software 5.0 (ThermoQuest Inc., San Jose, CA, USA) using an external standard method. All samples were prepared in triplicate, and the organic acid contents were expressed in mg/100 g bread or as mg/g sourdough (dry basis). Known concentrations of lactic acid, acetic acid, isobutyric acid, and butyric acid were used for calibration and verification of the eluted peaks.

### 2.10. Differential Scanning Calorimetry (DSC)

Bread staling was evaluated by recording the changes in starch retrogradation parameters upon storage of the bread samples at 25 °C using a DSC 3 calorimeter (Mettler-Toledo GmbH, Analytical, Zurich, Switzerland); bread crumb specimens were analyzed at two different time intervals for CB (2 and 96 h) and three different time intervals for sourdough breads (2, 96, and 144 h). Portions of approximately 10–12 mg of lyophilized powders of bread crumb were mixed with distilled water at a ratio of ~ 30:70 *w*/*w*, placed in stainless steel pans, and sealed hermetically. After 2 h (allowing for proper particle hydration), the samples were heated from 5–120 °C at a heating rate of 5 °C/min. For equipment calibration, an indium sample was used, whereas an empty pan was used as a reference material. The estimated thermal parameters were the onset (T_o_), peak (T_p_), and endset (T_e_) temperatures of the starch retrogradation endotherm and the apparent melting enthalpy (ΔH_ret_) of retrograded amylopectin. For each bread formulation, three replicates were examined by DSC.

### 2.11. Shelf Life Estimated by Mold Growth

The spoilage caused by mold growth was assessed by visual inspection. Slices of 3 cm thickness were cut from the center of the breads, packed in polyethylene bags, and incubated at 25 °C for 18 days. The area covered by mold growth was determined by image analysis using ImageJ 1.41o software and expressed as percent coverage of the bread slices by mold.

### 2.12. FTIR Spectroscopy Analysis

Fourier transform infrared (FTIR) spectroscopy was used to determine any changes occurring in the secondary protein structure of doughs prepared with different leavening agents before and after fermentation and in the crumb of the final baked bread. Dough samples and final bread crumb samples were placed on the ATR sampling accessory MIRacle ™-Universal ATR (Pike Technologies, Madison, WI, USA) with a 3-Reflection Diamond/ZnSe Performance Crystal Plate, and constant pressure was applied by the pressure tool. The samples were scanned (32 scans) with an FTIR spectrometer (FTIR 6700 series, JASCO, Tokyo, Japan) at a resolution of 4.0 cm^−1^, in the area of 4000–650 cm^−1^, in triplicate. Manipulation and processing of the spectra were performed according to the description given by Kotsiou et al. [30], using the Spectramanager V.2.15.15, Jasco, and SpectraGryph v.1.2.13 software (F. Menges Spectragryph–optical spectroscopy software, Oberstdorf, Germany). The spectra of dough and bread crumbs were further analyzed by the Gaussian curve fitting model (R^2^ > 98%), using the MagicPlot Student 2.9.3 free software (Magicplot Systems, LLC, Saint Petersburg, Russia), and applied to the Amide III region of the spectra (1200–1350 cm^−1^) [31]. The specific secondary structures were identified first by applying the second derivative deconvolution procedure with SpectraGryph v.1.2.13. The contribution of each structure obtained was expressed as % area of the total area of the absorption in the Amide III region. The areas of the obtained peak components were assigned to specific secondary conformations; i.e., the region of 1215–1254 cm^−1^ corresponds to *β*-sheet conformation, the region of 1254–1278 cm^−1^ corresponds to random structures, the region of 1278–1295 cm^−1^ corresponds to the *β*-turn conformation, and the region of 1295–1350 cm^−1^ corresponds to the *α*-helix [31,32], while an intense absorption peak at 1206 cm^−1^ corresponds to the tyrosine residue [33]. FTIR intensities of crumb samples from fresh (day 0) and stored breads for 4 or 6 days at 25 °C were also reported for the region between 945 and 1066 cm^−1^, where the characteristic bands of the flour starch are located.

### 2.13. Sensory Analysis of Breads

Sensory analysis was conducted by 10 trained panelists, male and female, aged 25–40 years old. All panelists frequently consumed bread at least once a week and were trained for the quantitative descriptive analysis test. Three sourdough bread samples, along with CB, used as a reference specimen, were assessed at two different time intervals of storage (1 and 4 days) at 25 °C using a 9-point scale according to their degree of similarity [34]. The sensory parameters that were evaluated for breads after one day of storage were: typical wheat bread aroma, non-wheat aroma, ‘butter-like’ aroma, typical wheat taste, non-wheat taste, ‘butter-like’ taste, sour taste, and bitter taste. For breads after 4 days of storage, the evaluated parameters were bread staling and ‘butter-like’ aroma. The overall acceptability was also evaluated by 75 non-trained panelists using a scale from 1 to 9 [35]. All sensory parameters, their descriptors, and definitions are provided in Appendix A.

### 2.14. Statistical Analysis

All the results are expressed as mean values of three independent experiments ±SD and comparisons were made using IBM SPSS statistical software (version 22.0, IBM Corp., Armonk, NY, USA), through ANOVA (Tukey’s test), at a significance level of *p* = 0.05. Mean values of % areas and peak intensities of the processed FTIR spectrum were compared by ANOVA, combined with the Tukey test and *t*-test, using SPSS Statistics 27.0.1.0 to determine any significance of differences among samples.

## 3. Results and Discussions

### 3.1. Identification of Clostridia from Chickpea Sourdough

It has been demonstrated that the microflora of fermented chickpeas consists mainly of bacilli (*Bacillus cereus*, *Bacillus licheniformis*, *Bacillus thuringiensis*) [21,36] and clostridia (*Clostridium perfringens*, *Clostridium beijerinckii*, *Clostridium sartagoforme*) [21,36,37,38]. In the present study, the dominant microorganisms found after chickpea sourdough fermentation were also clostridia (specifically, *C. perfringens* strains), reaching counts of 7.04 ± 0.20 log cfu/g, as characterized by SDS-PAGE, 16S rDNA sequence analysis, and toxin-typed strain identification. The SDS-PAGE of the whole cell protein fraction indicated that all isolates had identical protein patterns to a reference strain, *C. perfringens* DSM 756. This assignment was further confirmed by 16S rDNA PCR; amplicons sequenced and compared with those in the GenBank using the BLASTN algorithm showed homologies >99% found with deposited *C. perfringens* strains. Multiplex PCR with the four sets of primers for α, β, ε, and CPE toxin provided one single amplified product (324 bp), corresponding to toxin α, thus indicating that all isolates belonged to toxin type A and lacked the cpe gene. Additionally, neither the use of a second specific primer revealed the presence of the cpe toxin gene. The above results clearly indicate that indigenous strains of *C. perfringens* from traditional chickpea sourdough fermentation belong to toxinotype A and do not carry the cpe gene, thus being unable to produce enterotoxin. Moreover, according to Hatzikamari et al. [21], no toxin formation was detected in the same type of sourdough fermented by indigenous *C. perfringens* strains when the product was tested in mice. Wild strains of *C. perfringens* are almost always enterotoxin-negative [39], and it is considered that only strains that are heated repetitively, mostly present in a kitchen environment, are enterotoxin-positive [40]. Even in the case of CPE-positive strains encountered in chickpea sourdough that can produce enterotoxin, this would be inactivated upon subsequent bread baking, as this toxin is a heat-labile protein that is destroyed by heating at 60 °C for 5 min [41]. Consequently, there is no concern for any health hazard after consumption of breads made with chickpea sourdough preparations, considering that the temperature in the center of the bread crumb reaches ~96 °C at the end of baking.

### 3.2. Sourdough with Clostridia as a Starter Culture

The strain *C. perfringens* CP8 isolated from the autochthonous microflora of chickpea sourdough was finally used as a starter culture and inoculated into chickpea flour. After 2 h of fermentation at 37 °C, the initial population (6.62 ± 0.04 log cfu/g) increased significantly (*p* < 0.05) by 2.53 log counts (Figure 2a). With the addition of wheat flour and incubation for one more hour, the bacterial population remained relatively constant (8.88 ± 0.09 log cfu/g). Moreover, the initial pH value of 6.56 ± 0.14 decreased to 5.52 ± 0.13 in the first two hours of chickpea flour fermentation and then decreased further (5.03 ± 0.06) by the end of fermentation (Figure 2b). Similarly, the TTA increased from 1.17 ± 0.15 mL of 0.1 M NaOH to 3.53 ± 0.38 mL of 0.1 M NaOH in the first two hours of fermentation, and after the addition of flour, it increased at a faster rate, reaching 6.53 ± 0.45 mL of 0.1 M NaOH.

### 3.3. Drying of Liquid Sourdough (LS)

The drying of sourdoughs may affect the viability of the microorganisms, as demonstrated in previous studies, where a decline in the populations of lactic acid bacteria and yeasts was observed [42,43]. Thus, before drying, specific low molecular weight carbohydrates (i.e., maltodextrin, trehalose, and sucrose) were added to the LS as drying aids, either alone or in combinations (Appendix A), to protect *C. perfringens* CP8 cells from damage during the drying process. A decline in viability of at least 3 log cfu/g was noted by both drying methods employed to dry the sourdough preparations (Appendix A). In general, for all tested drying treatments, the inclusion of maltodextrin and/or trehalose seemed to exert a protective action for cell viability, and thus the viability of the starter was maintained at >4 log counts (Appendix A). Instead, the addition of sucrose had an adverse effect on viability, with the FDSS exhibiting the lowest population (3.25 ± 0.21 log cfu/g). In oven-dried and freeze-dried samples when extra wheat flour was added, the viability was boosted in the presence of sucrose only in the case that maltodextrin was also incorporated into sourdough, while the samples containing only sucrose exhibited the lowest population counts (~3.4 log counts), indicating that this sugar itself had no positive effect during drying. It is noteworthy that among all preparations, the highest bacterial population (5.24 ± 0.14 log cfu/g) was found for the FDSM sourdough when the sourdough was dried by freeze-drying without adding extra wheat flour before treatment.

The protective effect of low molecular weight carbohydrates on cells upon freezing and dehydration is generally thought to be related to the stabilization of proteins and other cellular biomolecules (e.g., membrane lipid bilayers) by restructuring the water molecules around the carbohydrate molecules and increasing the compactness of the three-dimensional structure of other biomolecules (smaller hydrodynamic volume) and organized cellular assemblies [44,45,46]. Maltodextrins are starch hydrolyzates that have also the ability to bind water and penetrate the cell membranes (transported into the bacterial cytoplasm), and these functions are enhanced as their molecular weight decreases [47,48]. According to vitrification theory, disaccharides and oligosaccharides may form high-viscosity glasses upon cooling of their aqueous solutions due to the freeze-concentration of these solutes [49,50]. In the viscous glass, water and bacteria are ‘immobilized’. Overall, the low mobility is likely to be responsible for the prevention of deteriorative reactions like protein unfolding and/or damage caused by large crystals during the freezing stage of the chickpea sourdough preparations prior to freeze-drying.

### 3.4. Storage Stability of Freeze-Dried Sourdoughs

Based on the above findings, the freeze-dried sourdough with maltodextrin (FDSM; showing the highest viability of *C. perfringens* CP8) and the sourdough without any carbohydrate addition (FDS) were stored at room temperature, in hermetically sealed polyethylene bags under vacuum in order to monitor the culture viability for 5 months (Figure 3a).

The results showed that the initial log counts in FDS (4.03 ± 0.18 logcfu/g) and in FDSM (5.24 ± 0.14 logcfu/g) were maintained at the same levels for 3 months and 2 months, respectively, and decreased gradually thereafter, by 1.68 and 1.79 log counts, until the end of the 5-month storage period. It is thus concluded that freeze-dried sourdough, with or without the addition of maltodextrin, could be used as a leavening agent for baked goods for three months of storage in a lyophilized form.

### 3.5. C. Perfringens CP8 Growth during Breadmaking

Three bread formulations were produced using LS, FDS, and FDSM as leavening agents (Table 2). The count values of *C. perfringens* CP8, determined in the respective wheat doughs fortified with the chickpea sourdough preparations before and after 2 h of proofing, are presented in Figure 3b. Although the initial counts before proofing in the doughs with FDS and FDMS (3.4 ± 0.08 and 2.92 ± 0.06 log cfu/g, respectively) were significantly lower (*p* < 0.05) compared to those with LS (4.62 ± 0.12 log cfu/g), after proofing for 2 h, the populations exceeded the value of 5 log cfu/g, in all three fortified (with the chickpea sourdough) dough preparations, recording counts of 5.26 ± 0.20 log cfu/g, 5.46 ± 0.12, and 5.78 ± 0.06 for LS, FDS, and FDSM, respectively. It is thus concluded that cells from freeze-dried sourdoughs could be effectively activated and proliferate in the fortified wheat flour dough matrix upon dough proofing.

### 3.6. Organic Acids in Sourdough and Bread

*C. perfringens* strains are known to produce butyric acid by carbohydrate fermentation as one of their main metabolic products, along with lactic and acetic acids, as well as some amounts of isobutyrate depending on substrate composition and strain used [51,52,53]. Short-chain fatty acids such as acetate and butyrate may also originate from amino acid biosynthetic pathways, whereas branched-chain fatty acids like isobutyrate, isovalerate, and 2-methyl-butyrate can be formed by the catabolism of branched-chain amino acids such as leucine and valine [54]. In the present study (Table 3), acetic acid was the dominant acid detected in LS, FDS, and FDSM (39.05, 32.83, and 55.64 mg/mL, respectively), whereas lactic acid seemed to be of rather minor concentration. Moreover, both butyric and isobutyric acids, which are responsible for the characteristic ‘butter-like’ flavor in this type of sourdough, were present and found in the sourdough-fortified bread loaves as well (Figure 4). The highest concentration of isobutyrate produced by *C. perfringens* CP8 was detected in the FDSM chickpea sourdough preparation (10.55 ± 1.27 mg/g) (Table 3).

In all breads made with FDS and FDSM, the dominant organic acid was also acetic acid, followed by isobutyric and butyric acids, with lactic acid being detected in much lower concentrations (Figure 4). Butyric and isobutyric acids were not detected as expected in CB. Although yeast can produce acetic acid, its concentration was much lower in CB compared to breads made with LS, FDS, and FDSM; apparently, *S. cerevisiae* has the ability to produce acetic acid through aerobic and/or anaerobic conditions by degradation of flour fructans [55,56]. Following storage for 4 days, the organic acids remained at the same levels in some cases of the freshly prepared products (Figure 4); some minor increases in concentration noted during storage of the loaves were most likely due to moisture losses in the bread crumb since all concentrations were expressed as mg/100 g bread as is.

Overall, it seems that the concentration of specific organic acids is affected by the type of sourdough employed in breadmaking. Regarding the lyophilized sourdoughs, higher amounts of individual and total organic acids were observed in the case of the FDSM preparation (Table 3). In breads, the profile of organic acids produced also differs among the samples leavened by different types of sourdough, with BLS and BFDS exhibiting higher levels of acetic and isobutyric acids, and BFDS and BFDSM being enriched in butyric acid (Figure 4).

### 3.7. Loaf-Specific Volume

The specific volume of breads leavened by LS, FDS, and FDSM sourdoughs was examined in comparison to CB leavened by baker’s yeast (Table 4); the loaf-specific volume increased with the addition of all sourdoughs, with BFDS showing the highest specific volume (3.2 mL/g). The increase in loaf volume may be fostered by enhanced gas production and the enzymatic activity of the bacterial culture present in the sourdough. In fact, the presence of α-galactosidase, cellulase, invertase, proteinase, and amylase activities in fermented chickpeas has been reported [17].

An increase in bread’s specific volume was also found by Gül et al. [57] and Sayaslan et al. [58] in wheat breads made with fermented-chickpea sourdough preparations; similar findings have also been reported for gluten-free breads made by both liquid and freeze-dried sourdoughs in which the autochthonous microorganisms from fermented chickpea were used as a starter [19,20]. In another study on gluten-free breads, the loaf volume increased with the addition of maltodextrins with DE 15.3–21.8, whereas the addition of a maltodextrin with DE 3.6 (higher molecular weight) had the opposite effect [59]. This could be explained by the increased concentration of fermentable low-molecular-weight carbohydrates in the case of higher DE maltodextrins [60] and the enhanced viscosity of the doughs when fortified with high-molecular-weight starch hydrolysate. In the present study, the BFDS exhibited the highest loaf volume among all the bread formulations (Table 4), indicating that the maltodextrin-containing freeze-dried sourdough (FDSM) had a negative effect on specific volume in comparison with the FDS preparation, although in comparison to the BLS there were no significant differences (*p* < 0.05).

The specific volume of wheat breads reflects the ability of dough to expand and maintain gases during proofing and the early stages of baking [61]; it is linked with flour protein quality (gluten protein fractions), recipe formulation, gluten network development (continuity) upon mixing of all dough ingredients, and the fermentation conditions [62,63]. Sourdough breads have a higher volume than breads leavened by baker’s yeast due to the ability of the composite gluten-starch matrix to allow gas cell expansion and restraining the losses of gases in the acidic doughs prior to the heat setting of the hydrated protein-starch composite network structure [64] as well as the increased solubility of pentosans following fermentation [65]. As supported by literature findings, the use of sourdough in breadmaking is an effective way to increase bread volume [66,67].

### 3.8. Color Parameters

The color of bread also changes during the baking of wheat dough as the crust is formed. These changes are probably caused by sugar caramelization and the Maillard reaction products [68,69,70,71]. The most important color parameter is L*, since it reflects crust brightness [68,72,73]. The sourdough-fortified breads generally exhibited lower L* values, indicating a darker crust color, in comparison to CB (Table 4). All sourdough breads had a higher a* value in comparison to CB, indicating a greater red color component. These results are in agreement with findings reported by Ozulku and Arizi [74], who also observed a reduction in L* value and an increase in the a* value of breads fermented with a sourdough containing chickpea flour. According to the b* value, BLS and BFDSM samples were similar to the control (Table 4) and in agreement with the data of Ertop and İbrahim Tuğkan et al. [75], as they found no significant differences in the b* values of the crust between breads leavened by baker’s yeast and breads made with the addition of a dried chickpea sourdough. In contrast, the BFDS exhibited a lower b* value compared to control bread (Table 4). Sayaslan and Şahin [58] also reported reduced yellowness in the crust of breads leavened by fermented-chickpea sourdough.

### 3.9. Moisture Content of Breads

During storage, bread loses its freshness, and the bread crumb becomes firmer due to moisture loss (water migration from crumb to crust), the retrogradation of starch molecules, and the transformation of the hydrated gluten matrix into a glassy state [76]. The bread crumb texture is one of the most important quality attributes of baked products, with the amount of water present in the crumb greatly affecting their textural characteristics [77]. The evolution of moisture content in the crumb and crust of all bread samples upon storage is presented in Table 5. The evolution of moisture content changes indicates an increase in water content in the crust along with a decline in water content in the crumb samples, complying with the well-known process of water migration from crumb to crust during storage of many baked products.

Overall, the incorporation of chickpea sourdough into the wheat flour dough seems to affect the moisture redistribution between crumb and crust during storage of the sourdough-fortified wheat breads. After 4 days of storage (at 25 °C), CB and BFDS exhibited the highest crumb moisture loss (>2%), whereas following 6 days of storage, among the sourdough-fortified breads, BFDSM showed the highest moisture loss (approximately 4%). On the other hand, BLS exhibited the highest crumb moisture content among all tested breads throughout the entire period of storage (Table 5), which can be ascribed to the ability of sourdough to increase the solubility of dietary fibers present in the wheat and chickpea flours and hence their water holding capacity. Concerning the crust moisture content, the CB sample exhibited the highest increase (12.5%) of moisture, while BLS, BFDS, and BFDSM showed smaller increases (<11.7%) after 4 days of storage (Table 5). The lowest crust moisture content, which is related to the appealing sensory attribute of crust crispness, was noted for the BFDS sample throughout the entire storage period.

### 3.10. Texture Analysis of Breads

Texture properties (crust and crumb hardness, cohesiveness, springiness, and chewiness) in breads during storage at 25 °C are presented in Figure 5. Crumb hardness gradually increased during storage, while crust hardness was reduced. CB had a harder crumb than that of sourdough-fortified breads. Among the different sourdough breads, the crumb hardness values were higher in the FDSM bread (Figure 5a), which could be partially ascribed to the low crumb moisture content of this product (Table 5). Similar results have been reported on whole [67] and white [66] wheat breads leavened by fermented chickpea sourdough as well as gluten-free breads made by liquid or freeze-dried sourdough in which the autochthonous microflora from fermented chickpea was used as a starter [19,20]. The positive effect on crumb softness might be due to the increased enzymatic activity, such as bacterial proteinase and α-amylase [17], that may cause partial gluten and starch hydrolysis and thereby lead to interruption of the continuous starch-protein matrix. This might also contribute to the higher specific volumes noted for the sourdough breads compared to the control (Table 4), as previously suggested by other authors [19,20,78,79]; high loaf volumes can also contribute to the crumb softness (open crumb structure). The crust hardness was similar for all sourdough-fortified breads, but lower than CB throughout the entire storage period (Figure 5b). Chewiness values (Figure 5c) of the bread crumb were also significantly lower in the sourdough bread samples. During bread staling, crust softening is noted due to the migration of water from crumb to crust, whereas crumb hardening is also due to retrogradation of starch and water displacement among the components at a molecular level. In fact, further to water loss, crumb firmness may increase due to the transition of gluten from a rubbery (water-plasticized) to a glassy state caused by water transfer from gluten to the starch component [80,81,82].

After the application of compression forces, the recovery ability of crumbs is expressed by springiness, which reflects crumb elasticity, while low cohesiveness values are related to crumbly bread. The results showed that CB and BFDSM had the lowest cohesiveness (Figure 5d) and springiness (Figure 5e), with no significant differences (*p* > 0.05) between them, whereas the BLS and BFDS breads were less friable and more elastic (higher cohesiveness and springiness values, respectively). During storage, the staling events may reduce the cohesiveness, probably due to the weakening of intermolecular associations between ingredients, leading to crumb crumbling, which is also correlated with water loss [83]. Indeed, positive linear correlations (*p* < 0.05) were noted between cohesiveness and crumb moisture content, throughout storage for all samples (CB: r = 0.869, *p* = 0.02, *n* = 9; BLS: r = 0.795, *p* = 0.02, *n* = 12; BFDS: r = 0.685, *p* = 0.014, *n* = 12; BFDSM: r = 0.689, *p* = 0.13, *n* = 12).

Overall, it seems that incorporation of both liquid and freeze-dried sourdoughs (with or without inclusion of maltodextrin in the chickpea sourdough) into the wheat bread formulations contributes to the extension of product shelf life since the fortified breads exhibited significantly lower hardness and chewiness as well as higher cohesiveness and springiness of the crumb than the control sample during storage at 25 °C (Figure 5). Similar conclusions were reached by Rizzello et al. [84], who showed that the addition of quinoa sourdough to white wheat bread positively affected the texture, volume, and sensory properties of the product. Additionally, it was reported that freeze-dried wheat germ sourdough incorporation into white bread [71] and the use of liquid or freeze-dried sourdough by employing a fermented chickpea extract as a starter in gluten-free breadmaking [19,20] also improved the texture and sensory characteristics of the end-products.

### 3.11. Starch Retrogradation in Bread Crumb

Bread staling data, as monitored by calorimetric measurements of bread crumb samples for the extent of amylopectin retrogradation during storage (i.e., melting enthalpy, ΔH_ret_, values), are summarized in Table 6. The endothermic peak observed around 48–53 °C, reflecting the melting of retrograded amylopectin, increased in all cases as expected during bread staling [76,85,86]. These changes in starch chain reordering point to another contributing factor to bread crumb hardening besides the impact of moisture loss and its redistribution among the bread components, i.e., the development of the crystalline molecular assemblies of amylopectin short chains [9,65,86,87,88,89].

No major changes in T_o_ (onset) and T_e_ (endset) of the melting endotherm for retrograded amylopectin were noted during storage of the bread samples. Instead, a slight increase in the T_p_ was observed, implying the progressive reorganization of the amylopectin crystalline structures and complying with the transient nature of the hydrated starch network structure as it gradually evolves [86,87,90,91].

Beneficial effects towards the reduction of bread staling events have been reported upon acidification and were attributed to metabolite products generated during fermentation [92,93], as a result of the amylolytic and proteolytic activities of microorganisms [94,95]. Additionally, the moisture content could contribute to the retardation of bread staling [96] and consequently to prolonging shelf life. The BLS and BFDS samples exhibited rather constant ΔH_ret_ values between 96 and 144 h of storage compared with the BFDSM bread, which continued to show increasing trends in amylopectin chain ordering (Table 6). In general, for the same storage time interval (96 h), BLS and BFDS breads exhibited a lower extent of starch retrogradation compared to the other two samples. These results indicate that the addition of sourdough could efficiently delay amylopectin recrystallization during storage and are in line with findings from previous studies [19,65,97,98].

### 3.12. Mold Growth in Breads

Surface coverage (%) of breads by mold growth during storage at 25 °C is presented in Figure 6. Usually, mold spoilage of bakery products is caused by *Penicillium* spp., *Aspergillus* spp., *Eurotium* spp., *Wallemia* spp., *Fusarium* spp., and *Cladosporium* spp. [99,100]. Compared to CB, which showed mold growth even on day 5, the BFDS and BLS samples exhibited the highest mold inhibition, as visual growth was noted after 11 and 10 days of storage, respectively, probably due to the higher concentration of acetic acid present in these breads (Figure 4), as reported by other researchers as well [52,101]. A relatively short time for mold growth (day 7 of storage) also occurred in the case of BFDSM (Figure 6), most likely due to the presence of maltodextrin, which can serve as an additional substrate for mold growth. Overall, it is obvious that the addition of sourdough to the wheat bread formulations not only prolonged their shelf life by delaying staling but also effectively delayed spoilage due to mold growth. The antifungal activity of sourdoughs is well documented in breads formulated with typical sourdoughs fermented by LAB as a starter culture [102] and has been attributed to the production of organic acids by LAB [103], mainly acetic and lactic acids [9,104,105,106].

### 3.13. FTIR Spectroscopy Analysis in Sourdough and Breads

Compared to the Amide I region, the Amide III region of proteins is more sensitive to changes occurring in secondary structures because it is dominated by the in-phase combination of N-H in-plane bending and C-N stretching vibrations. Overlapping of the secondary structure component bands is also limited in the case of the Amide III region; hence, the bands are better resolved. Moreover, the absorbance of water molecules in this region is insignificant [31,107]. The secondary conformational elements of the protein network developed during dough fermentation and bread baking were determined by applying the curve fitting method to the Amide III region.

#### 3.13.1. Protein Secondary Structure of Doughs

The protein network of bread doughs, regardless of the leavening agent used, consisted of *β*-sheet, random, *β*-turn, and *α*-helix conformations (Figure 7a–d). The peak constituent, centered at 1295 cm^−1^, yielded in all spectra was found on the border of *β*-turn and *α*-helix conformations. According to Cai and Singh [31], this band can be attributed to either *α*-helix or *β*-turn or a mixture of these two structures, i.e., a folding between these two formations. Jung et al. [33] assigned this absorption region to either *β*-turn or 3_10_
*α*-helix structure, i.e., an *α*-helix/*β*-turn intermediate protein conformation (called hereafter). Band deconvolution also yielded a band centered at 1206 cm^−1^, corresponding to exposed tyrosine residues on the surface of the protein network [33], which may interact with other components [108]. Table 7 presents the contribution of the different structural elements to the protein conformation of the doughs before and after proofing. In all bread dough samples (before and after proofing), the predominant secondary protein structure was the *β*-sheet structure (ranging from 32.79 to 39.85%), which is in accordance with previous findings [109,110], followed by an almost equal proportion of *α*-helix structures (ranging from 26.36 to 38.36%). The protein network of CB dough exhibited the highest composition in the *β*-sheet conformation (39.85%), compared to the doughs prepared with sourdough. Similar observations were reported by Zhang et al. [111] in a study about the effect of acid treatment on doughs. In general, the use of sourdough, regardless of its form, significantly influenced (*p* < 0.05) the protein structure of the dough, compared to the CB dough (Table 7). The inclusion of freeze-dried chickpea sourdough resulted in doughs with a lower (*p* < 0.05) content in the *β*-sheet structure (BFDS dough: 37.72%; BFDSM dough: 34.54%) compared to the content found in BLS dough (39.50%). Concerning the random structure, it was three times higher in CB (3.24%) compared to the dough formulations containing sourdough (BLS dough: 1.74, BFDS dough: 1.16 and BFDSM dough: 1.02%). However, in all dough samples, the contribution of random coil structural elements was the least among all determined secondary structures, which is in accordance with the findings of Nawrocka at al. [112]. The substitution of baker’s yeast with chickpea sourdough resulted in a significant increase (*p* < 0.05) in the *α*-helix structure (CB dough: 26.36%, BLS dough: 28.50%) (Table 7). Moreover, the incorporation of freeze-dried sourdough formulations in the wheat dough preparation resulted in a further increase in the *α*-helix structure proportion, up to 31.19 for BFDS and 35.86% for BFDSM. The BLS dough exhibited the lowest level (*p* < 0.05) of *β*-turn conformation (7.68%), compared to the highest observed in CB dough (9.11%, *p* < 0.05), followed by BFDS and BFDSM doughs (~8.45%, *p* > 0.05). Higher contents in *α*-helix and *β*-turn structures in doughs have been previously observed after weak acidification [111]. According to the “loop-train” model, the more flexible structures of *α*-helix and *β*-turn are mainly responsible for the loop region, while *β*-sheets form the “train” region. In acidic environments, *β*-sheets are repulsed resulting in “unzipped” forms that contain more flexible structures. However, Nutter et al. [113] pointed out that during sourdough incubation, the changes occurring in the secondary structure were dependent on the strain of bacteria employed and resulted in formation of *β*-sheet structures at the expense of *α*-helix conformation. Feng et al. [114] reported a positive correlation between *β*-sheet content and a negative correlation between *β*-turn content with dough viscosity. The high content of *β*-sheets has also been linked with limited gluten coherence and extendibility [111].

The substitution of baker’s yeast with liquid sourdough favored the formation of the *α*-helix/*β*-sheet intermediate structure from 14.95 to 16.22%, *p* < 0.05, while the inclusion of freeze-dried sourdough preparations resulted in its reduction (BFDS dough 14.76%, *p* > 0.05), especially when maltodextrin was included as a drying aid in the chickpea sourdough prior to its drying (BFDSM dough 13.57%, *p* < 0.05) (Table 7).

Generally, it is evident from the data presented in Table 7 that CB dough exhibited more profound changes in its protein secondary structures during fermentation (proofing stage), compared to the dough formulations containing sourdough as a leavening agent. A significant decrease in *β*-sheet structure ratios was observed in all samples, during fermentation of the doughs, which has been previously reported by Bock [110] for wheat flour-based bread dough systems. Instead, significant increases in *β*-sheet structures at the expense of *α*-helices were also noted during fermentation of wheat-rye sourdough made with lactic acid bacteria and were attributed to their acidic activity [113]. Similar trends were also observed in the secondary structure of bread dough with the inclusion of sourdough from fermented pitaya fruit with lactic acid bacteria [115]. The loss of *β*-sheet structure upon fermentation has been associated with dough structural relaxation processes [110]. Random conformational elements of the protein networks of CB and BLS doughs almost disappeared after fermentation, from 3.24 down to 0.25% for CB dough and from 1.74 down to 0.44% for BLS dough (*p* < 0.05), while the reduction in BFDS dough was less (from 1.16 to 0.63%) (Table 7). The BFDSM dough demonstrated the highest proportion of random structures (1.21%) and *α*-helix (38.36%) and the lowest proportion of *β*-turn structure (7.23%), composite (intermediate) structure (13.52%), and *β*-sheet structures (32.79%) amongst all leavened dough samples. A higher content of *β*-sheet structures in wheat doughs has been associated with lower dough quality and a poorer final bread loaf volume [109]. Indeed, the CB dough exhibited the highest content of *β*-sheet conformational elements (Table 7), in compliance with the lowest bread loaf volume of this bread preparation (Table 4).

#### 3.13.2. Protein Secondary Structure of Breads

Figure 7e shows the Amide III spectrum region of all bread samples, and Figure 7f shows the deconvoluted Amide III region after calculation of the 2nd derivative of the spectrum. Conformational changes in protein structure can be observed, especially in the region between 1325 and 1260 cm^−1^ upon baking (Figure 7e). The protein structure in all fresh bread samples (0D) was predominated equally by *β*-sheet (CB: 33.97%; BLS: 33.54%; BFDS: 36.35%; BFDSM: 32.56%) and *α*-helix structures (CB: 33.14%; BLS: 35.10; BFDS: 33.53%; BFDSM: 35.75%) (Table 8), which is comparable with the findings for wheat-flour-based breads [30,116]. It seems that CB and BLS lost more *β*-sheet structures (CB: −8.1%; BLS: −6.28% loss) during baking compared to the marginal changes noted for the BFDS and BFDSM samples (BFDS: +0.8% gain, BFDSM: −0.7% loss) (Table 7 and Table 8). This is partially contradictory to the increase in the *β*-sheet structure during baking in wheat-based bread reported by Bock et al. [109]. BFDSM lost *α*-helix structures during baking (−6.18% loss), while the protein networks of all the other samples gained *α*-helix conformational elements (CB: +19.81%; BLS: +7.63%; BFDS: +3.26%). A profoundly lower content (*p* < 0.05) of random structures was observed for BLS (1.86%), BFDS (1.51%), and BFDSM (2.22%) compared to CB (4.15%) (Table 8). The contribution of *β*-turn structure to the protein network conformation was around 5% (*p* < 0.05) for all fresh bread formulations.

A downward trend of *α*-helix during bread storage has been previously reported for wheat flour-based bread and wheat flour-based bread enriched with yellow split pea flour [30,116] and in the glutenin fraction of steamed bread [117]. The protein chains tend to get reorganized during storage as a response to changes in intermolecular interactions among water molecules, polysaccharides, and other proteins present in composite bread network matrices [118,119]. An increase in *β*-sheet structure has also been associated with gluten dehydration, occurring due to the antagonistic interactions between fibers/starch and gluten with water molecules or the effect of ethanol [109,120]. In the present study, the protein networks of BLS and BFDSM also revealed structural modifications during the storage of breads.

In most cases, the *β*-sheet and *β*-turn structures increased (*p* < 0.05) at the expense of the other protein structures after 6 days of storage (Table 8), partly due to the lower moisture content, particularly of the BFDSM after 6 d of storage (Table 5). Instead, the BFDS has not undergone major protein conformational changes during bread storage (Table 8), most likely due to the rather moderate water loss (Table 5). Overall, different moisture content losses during baking and storage among the bread samples (Table 5), along with variations in acidity, could be responsible for the observed conformational changes of the different protein networks in the stored breads.

#### 3.13.3. Starch Chain Conformation of Breads

Bands at 1047 and 1022 cm^−1^ have been associated with starch structures corresponding to molecularly ordered chain clusters (or crystalline structures) and amorphous regions, respectively [121,122,123]. Hence, the ratio R_1047/1022_ has been used by several researchers as an index of chain ordering and starch retrogradation [30,124]. In agreement with DSC measurements (Table 6, ΔH_ret_ values), the ratio R_1047/1022_ increased after storage in all bread samples (Table 8), as expected due to the staling process (implying transformation of amorphous chain domains towards more ordered structures in the starch molecules), i.e., a decrease in intensity at 1022 cm^−1^ along with an increase in intensity at 1044 cm^−1^ were noted [30,124,125]. The BLS and BFDS exhibited lower R_1047/1022_ rates of increase (+0.006 and +0.004, respectively; *p* < 0.05) during storage, reaching final values of R_1047/1022_ on the 6th day of storage equal to 0.718 and 0.717, respectively, which are significantly lower (*p* < 0.05) than those measured for CB and BFDSM (0.726 and 0.747, respectively; *p* < 0.05) (Table 8). 

The observed lower level of ordered starch structures of BLS and BFDS is, also, in the same vein with their lower hardness (Figure 5a) and ΔH_ret_ (Table 6) values of crumbs, as measured by TPA and DSC, respectively. The starch molecules of BFDSM exhibited the highest rate of transformation from amorphous to crystalline form and reached the highest final value of R_1047/1022_ on the 6th day of storage, compared to the other bread samples (Table 8). This observation is in agreement with the pronounced high content of *β*-sheet structures (45.5%) of this bread sample after 6 days of storage, which, as previously discussed, is linked with gluten dehydration [109,116] and therefore more available water molecules for promoting starch retrogradation and the shift of gluten from a rubbery to a glassy state; both these events contribute to a harder crumb texture.

### 3.14. Sensory Analysis

The BLS, BFDS, and BFDSM breads were subjected to sensory analysis and compared to CB leavened by baker’s yeast after 1 and 4 days of storage at room temperature (Figure 8a). Regarding aroma and taste, which are important sensory attributes of bread the evaluators detected lower typical ‘wheat type’ aroma and taste in all sourdough-fortified samples in comparison to CB. Additionally, higher scores for a ‘non-wheat type’ aroma and taste (without a specified identification) were assigned to the BLS and BFDS samples rather than the BFDSM sample. A ‘butter-like’ aroma, attributed to butyric and isobutyric acids, produced from the starter culture (*C. perfringens* CP8), was also perceived by the panelists in all sourdough breads, as has also been verified by HPLC analysis for butyric and isobutyric acid production in the chickpea sourdoughs (Table 3) and the respective bread samples (Figure 4). The evaluators have detected a ‘butter-like’ aroma and taste in all sourdough breads, with the freeze-dried sourdough inclusion resulting in the breads having the highest score (Figure 8a). A slightly sour taste was also detected for all bread samples, probably due to acetic and other organic acids produced (Figure 4). No bitter flavor notes have been reported (Figure 8a).

After 4 days of storage, ‘bread staling’ and the presence of a ‘butter-like’ aroma were assessed by the panelists (Figure 8a). The BLS and BFDS samples seemed to maintain their freshness longer than CB and BFDSM. The ‘butter-like’ aroma was also better preserved in the BLS and BFDS samples (5.30 and 5.00, respectively) after 4 days of storage. Overall, the chickpea sourdough-fortified breads, BLS and BFDS, maintained their freshness better than the control and BFDSM samples, which is in agreement with the findings obtained by TPA (Figure 5) and DSC (Table 6) analysis. Concerning the overall acceptability scores (Figure 8b), all sourdough breads received greater acceptability scores by the panelists in comparison to the CB sample; among the evaluated four breads, BFDS exhibited the highest mean score (6.30).

The findings of this study indicate that the addition of chickpea sourdough preparations offers a distinct ‘butter-like’ aroma and taste to wheat breads, which are perceived as pleasant by the panelists, increasing their overall acceptability. These results are in accordance with the findings of Gül et al. [57], who postulated that the addition of a chickpea-based leavening extract in bread formulations increases the acceptability of the fortified baked products. Similar results have also been reported by Xiao et al. [126], which showed that the addition of *Cordyceps militaris* (an endogenous fungus widely used in Asia as a medicinal fungus)-fermented chickpea flour in bread improves the organoleptic properties of wheat bread. Additionally, the preference for rice-based gluten-free bread was largely improved when liquid or freeze-dried sourdough made by an autochthonous fermented chickpea starter was included in the formulation [20].

## 4. Conclusions

In the present study, a chickpea sourdough fermented by a starter culture involving a *C. perfringens* strain isolated from the autochthonous microflora of fermented chickpea flour has been investigated as a dough improver in making wheat bread. *C. perfringens* has been identified as the dominant microbial species in traditional chickpea sourdough. Genotyping of several isolated strains by molecular methods classified them as belonging to type A and being CPE enterotoxin-negative. A selected isolated strain (CP8) was then employed as a starter culture to standardize the chickpea sourdough fermentation. The sourdough preparation was dried in two different ways (oven-dried vs. freeze-dried) with or without the inclusion of some protective carbohydrates in the fermented dough before drying; apparently, maltodextrin with DE of 17–19 was found to be an effective drying aid for preserving the viability of *C. perfringens* CP8 during drying. The chickpea sourdough in three forms (liquid, LS, and freeze-dried without maltodextrin, FDS, or with maltodextrin, FDSM) was evaluated as a dough improver/leavening agent to produce wheat breads, and the physicochemical properties of the breads were assessed. Inclusion of chickpea sourdough (LS or FDS) into wheat doughs had beneficial effects on the breads produced. The crust and crumb hardness were lower for all sourdough-fortified breads compared to the CB (control, without sourdough) sample, consistent with delayed staling events, as revealed by texture profile analysis (TPA) and differential scanning calorimetry (DSC). Moreover, the incorporation of LS and FDS in the fortified dough preparations delayed mold growth in breads for at least 9 days.

Organic acid analysis of the end products showed the presence of butyric and isobutyric acids, which deliver a pleasant ‘butter-like’ taste and aroma due to the fermentation of *C. perfringens* CP8 in the dough proofing stage and are detectable by sensory analysis. This characteristic aroma note was maintained in the fortified breads even after four days of storage at 25 °C.

Overall, the addition of chickpea sourdough fermented by *C. perfringens* CP8 as a starter culture in wheat breads improved their quality attributes. Additionally, the use of a freeze-dried sourdough preparation in breads exhibited similar and, in some cases, better functionality in comparison to its liquid counterpart. Thus, freeze drying could be a promising method for a ready-to-use chickpea sourdough preparation to standardize the production of traditional bread and other ‘clean label’ baked products with improved sensorial characteristics and shelf life.

## Figures and Tables

**Figure 1 foods-12-03112-f001:**
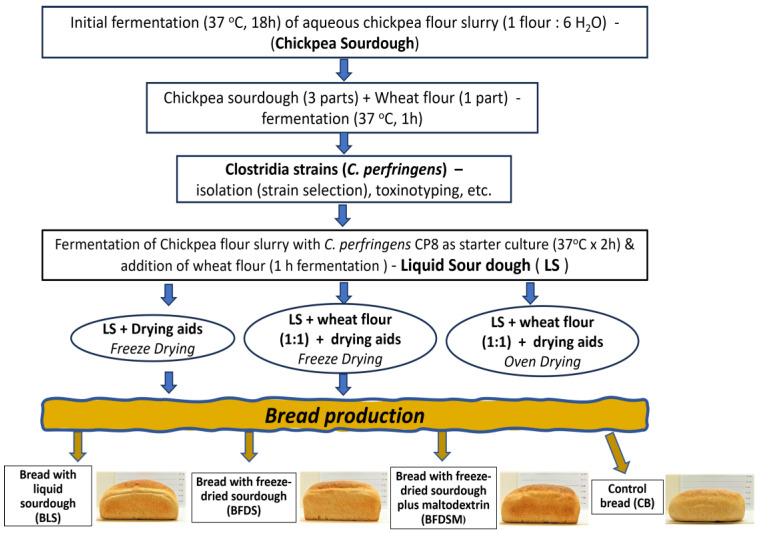
Flowchart illustrating the process of sourdough production and the production of bread formulations.

**Figure 2 foods-12-03112-f002:**
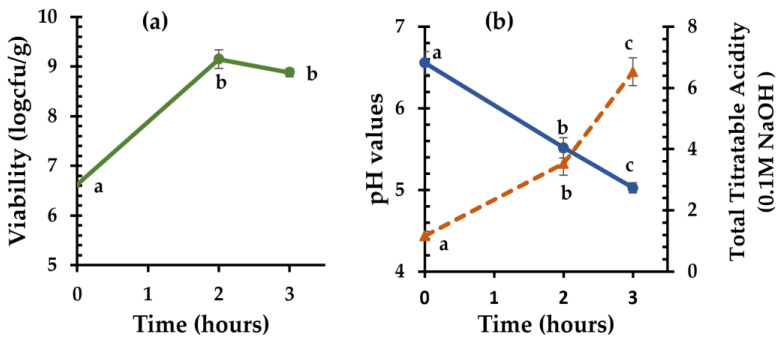
Cell viability of *C. perfringens* CP8 (**a**) and Total Titratable Acidity (dotted line) and pH values (solid line) (**b**) during chickpea sourdough fermentation at 37 °C for 2 h, followed by addition of wheat flour to the sourdough and fermentation at 37 °C for an additional 1 h. Mean values for the same parameter with different letters indicate significant differences (*p* < 0.05).

**Figure 3 foods-12-03112-f003:**
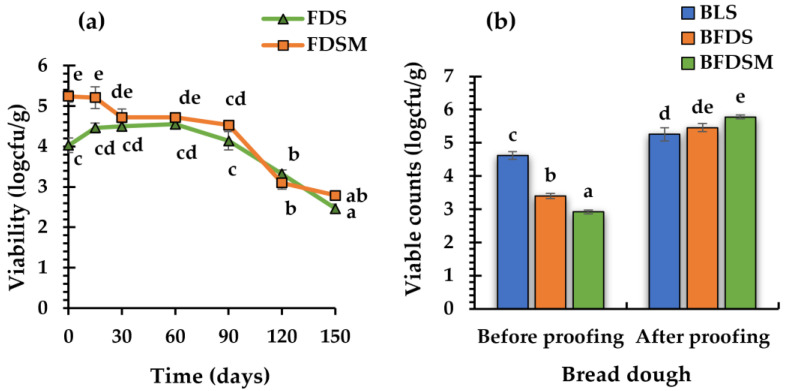
Viability profiles of *C. perfringens* CP8 during storage (25 °C) under anaerobic conditions (stored for 150 days) in freeze-dried chickpea sourdoughs, with or without the addition of maltodextrin in the preparations (**a**). *C. perfringens* CP8 viability in bread doughs prepared with addition in the formulations of liquid or freeze-dried sourdoughs, made with or without the inclusion of maltodextrin in the sourdough before drying, prior to proofing (immediately after ingredient mixing), and after 2 h proofing (prior to baking) (**b**). Bars represent the mean values of three replicates; mean values with different letters indicate significant differences (*p* < 0.05). Notation of samples as in Table 2.

**Figure 4 foods-12-03112-f004:**
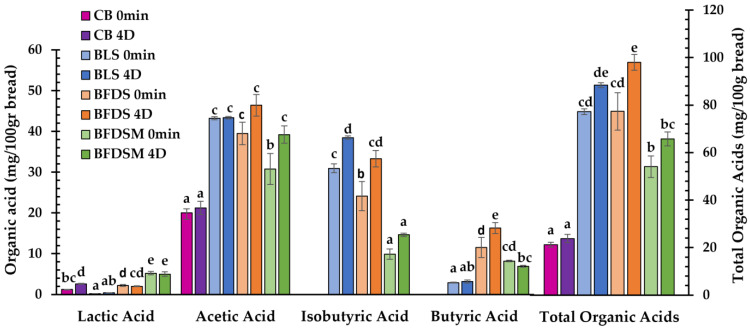
Concentration of lactic, acetic, isobutyric, and butyric acids and total organic acids (mg/100 g bread) in the crumb of wheat bread leavened by baker’s yeast (CB) and breads formulated with the addition of liquid (BLS), freeze-dried (BFDS), and freeze-dried fortified with maltodextrin (BFDSM) sourdoughs at 0 min and after 4 days (4D) storage at 25 °C. Mean values with different letters for the same organic acid indicate significant differences (*p* < 0.05).

**Figure 5 foods-12-03112-f005:**
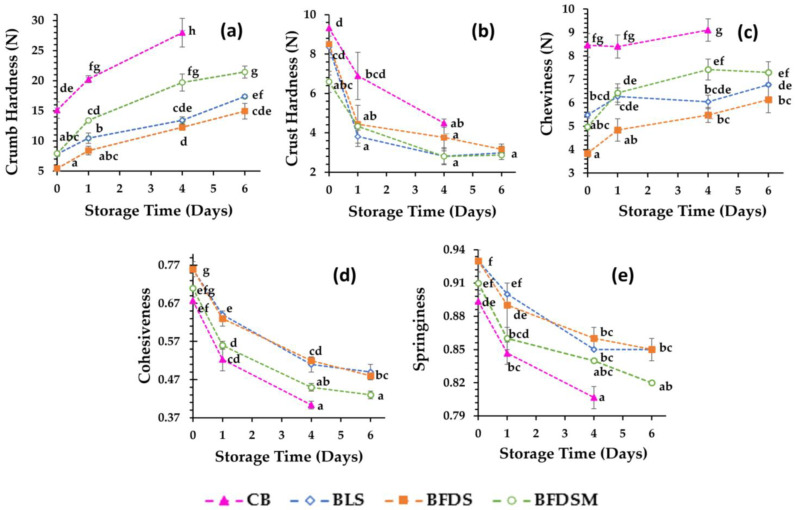
Evolution of crumb hardness (**a**), crust hardness (**b**), crumb chewiness (**c**), crumb cohesiveness (**d**), and crumb springiness (**e**) during storage at 25 °C. Mean values with different letters indicate significant differences (*p* < 0.05). Notation of samples as in Table 2.

**Figure 6 foods-12-03112-f006:**
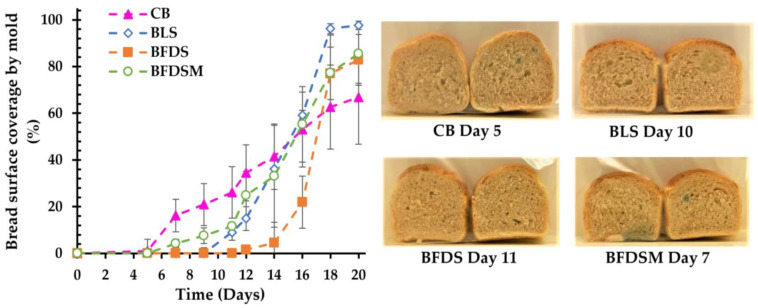
Surface coverage by mold in breads fermented by baker’s yeast and chickpea sourdoughs during 20 days of storage at 25 °C and representative images of center slices at the time that visual mold growth was noted. Notation of samples as in Table 2.

**Figure 7 foods-12-03112-f007:**
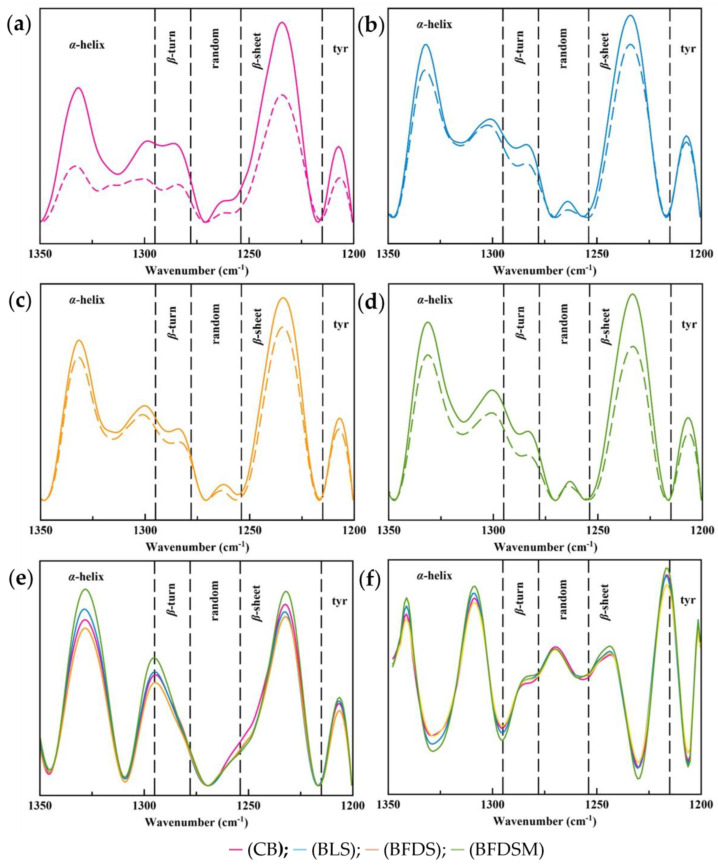
FTIR spectra of the Amide III region of doughs (**a**–**d**) prior to proofing (immediately after ingredient mixing) (solid line) and after 2 h of proofing (prior to baking) (dotted line), fresh breads (**e**) and the second derivative of the Amide III region of fresh breads (**f**); Notation of samples as in Table 2.

**Figure 8 foods-12-03112-f008:**
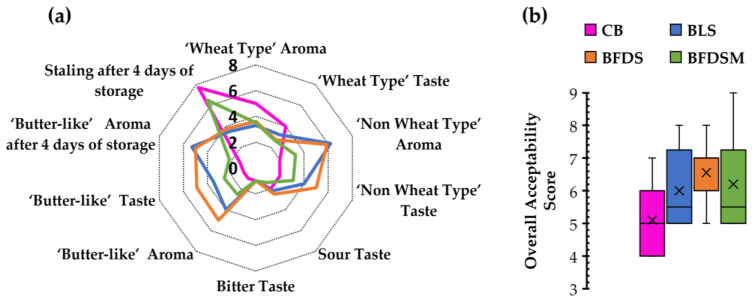
Sensorial characteristics of breads fermented by baker’s yeast and by chickpea sourdoughs to evaluate taste and aroma (after 1 day of storage), staling and ‘butter-like’ aroma (following 4 days of storage at 25 °C) (**a**); and overall acceptability (after 1 day of storage) (**b**). Notation of samples as in Table 2.

**Table 1 foods-12-03112-t001:** Primers for toxinotyping of *C. perfringens*.

Toxin	Sequence 5′-3′	Amplicon (bp)
α	GCTAATGTTACTGCCGTTGA	324
	CCTCTGATACATCGTGTAAG	
β	GCGAATATGCTGAATCATCTA	196
	GCAGGAACATTAGTATATCTTC	
ε	GCGGTGATATCCATCTATTC	655
	CCACTTACTTGTCCTACTAAC	
CPE	GGAGATGGTTGGATATTAGG	233
	GGACCAGCAGTTGTAGATA	
CPE	GGGGAACCCTCAGTAGTTTCA	506
	ACCAGCTGGATTTGAGTTTAATG	

**Table 2 foods-12-03112-t002:** Bread formulations with the addition of liquid (BLS), freeze-dried (BFDS), freeze-dried sourdough fortified with maltodextrin (BFDSM), and control bread (CB) made solely with baker’s yeast.

Component (g)	CB	ΒLS	BFDS	BFDSM
Wheat flour	92.50	75.00	75.00	75.00
Chickpea flour	7.50	-	-	-
Water	59.00	13.50	59.00	59.00
Baker’s yeast	1.00	0.20	0.20	0.20
Liquid sourdough (g H_2_O + dry solids)	-	(45.50 + 25.00)	-	-
Dried FDS ^1^	-	-	25.00	-
Dried FDSM ^1^	-	-	-	25.00
Salt	1.00	1.00	1.00	1.00
Vinegar	1.50	1.50	1.50	1.50

^1^ FDS: Freeze-dried sourdough; FDSM: Freeze-dried sourdough with the addition of 15% (on LS basis) maltodextrin.

**Table 3 foods-12-03112-t003:** Organic acids of liquid chickpea sourdough after fermentation at 37 °C for 3 h and after freeze drying with or without the addition of maltodextrin.

	Organic Acids (mg/g Sourdough)			
	Lactic Acid	Acetic Acid	Isobutyric Acid	Butyric Acid
LS ^1^	4.18 ± 0.27 a ^2^	39.05 ± 0.06 a	6.54 ± 0.71 a	4.69 ± 0.59 a
FDS	4.24 ± 0.06 a	32.83 ± 0.21 a	6.90 ± 0.32 a	13.58 ± 0.29 b
FDSM	4.41 ± 0.22 a	55.64 ± 6.07 b	10.55 ± 1.27 b	31.39 ± 3.41 c

^1^ LS: liquid sourdough; FDS: freeze-dried sourdough; FDSM: freeze-dried sourdough fortified with maltodextrin. ^2^ Mean values with different letters in the same column indicate significant differences (*p* < 0.05).

**Table 4 foods-12-03112-t004:** Loaf-specific volume and crust color parameters of control and sourdough breads.

	CB ^1^	BLS	BFDS	BFDSM
Loaf-specific volume (mL/g)	2.20 ± 0.04 a ^2^	2.64 ± 0.05 b	3.19 ± 0.08 c	2.69 ± 0.11 b
Crust color parameters				
L*	61.07 ± 1.22 c	50.82 ± 0.81 a	56.35 ± 1.33 b	50.62 ± 1.26 a
a*	8.91 ± 0.28 a	12.27 ± 0.05 b	14.45 ± 0.01 d	13.58 ± 0.44 c
b*	33.30 ± 1.23 b	28.22 ± 0.28 ab	27.43 ± 0.96 a	29.50 ± 0.35 ab

^1^ Notation of samples as in Table 2. ^2^ Μean values with the same letter in the same row indicate non-significant differences (*p* > 0.05).

**Table 5 foods-12-03112-t005:** Changes in crumb and crust moisture content in breads made by baker’s yeast and by the addition of chickpea sourdough preparations at 0, 1, 4, and 6 days of storage at 25 °C.

Bread Moisture (%)					
		0 Day	1 Day	4 Day	6 Day
**Crust**	CB ^1^	17.84 ± 0.49 a ^2^	27.23 ± 0.11 b	30.38 ± 0.69 c	
	BLS	20.60 ± 0.26 a	27.34 ± 0.21 b	31.25 ± 0.60 c	32.69 ± 0.84 c
	BFDS	16.19 ± 1.55 a	22.31± 0.14 b	27.85 ± 0.29 c	26.64 ± 0.70 c
	BFDSM	20.32 ± 0.45 a	21.86 ± 0.38 a	31.56 ± 0.60 b	30.78 ± 1.23 b
**Crumb**	CB	45.02 ± 0.59 a	44.04 ± 0.38 ab	42.78 ± 0.64 b	
	BLS	46.08 ± 0.37 a	45.96 ± 0.54 ab	44.83 ± 0.62 b	43.33 ± 1.81 b
	BFDS	42.84 ± 0.18 a	42.67 ± 0.57 b	40.34 ± 0.11 b	41.87 ± 0.06 b
	BFDSM	42.84 ± 0.94 a	42.68 ± 0.03 ab	41.10 ± 1.02 b	38.77 ± 0.98 c

^1^ Notation of samples as in Table 2. ^2^ Mean values with the same letter in each row indicate no significant differences (*p* > 0.05).

**Table 6 foods-12-03112-t006:** Evolution of amylopectin retrogradation in crumbs for breads leavened by baker’s yeast and by three different types of sourdough (liquid and freeze-dried preparations with or without inclusion of maltodextrin in the chickpea sourdough prior to drying), in aqueous dispersion (bread crumb/water 30:70 *w/w*) during storage at 25 °C for 144 h.

	Melting Temperature of Retrogradation Endotherm			Retrogradation Melting Enthalpy
Samples ^1^	Onset, T_o_ (°C)	Peak, T_p_ (°C)	Endset, T_e_ (°C)	ΔH_ret_, (mJ/mg Flour d.b.)
CB 2 h	43.35 b ^2^	48.02 a	60.18 b	0.13 a
CB 96 h	43.06 b	52.44 de	62.14 d	2.43 de
BLS 2 h	43.03 b	47.92 a	59.10 a	0.13 a
BLS 96 h	44.77 de	53.15 e	59.24 a	1.09 b
BLS 144 h	45.57 e	52.95 de	60.10 b	1.47 bc
BFDS 2 h	45.25 e	50.42 bc	60.55 b	0.12 a
BFDS 96 h	44.28 cd	52.22 de	60.35 b	1.55 bc
BFDS 144 h	43.81 bc	52.64 de	60.92 c	1.88 cd
BFDSM 2 h	43.07 b	50.11 b	60.35 b	0.29 a
BFDSM 96 h	41.70 a	51.87 cd	62.80 e	1.89 cd
BFDSM 144 h	43.25 b	53.14 e	63.415 f	2.53 e

^1^ Notation of samples as in Table 2. ^2^ Μean values with different letters in the same column are significantly different (*p* < 0.05).

**Table 7 foods-12-03112-t007:** Contribution of the protein secondary structures (%) in the spectral region of Amide III of bread doughs before and after dough proofing.

	CB ^1^ (% Area)		BLS (% Area)		BFDS (% Area)		BFDSM (% Area)	
Secondary Structure Assignment	Dough	Leavened Dough	Dough	Leavened Dough	Dough	Leavened Dough	Dough	Leavened Dough
Tyr ring	6.47 ± 0.06 a ^2^	7.13 ± 0.41 a	6.36 ± 0.35 a	6.98 ± 0.36 a	6.72 ± 0.21 a	6.89 ± 0.41 a	6.57 ± 2.16 a	6.89 ± 0.46 a
*β*-sheet	39.85 ± 0.09 f	36.96 ± 1.21 de	39.50 ± 2.01 f	35.79 ± 1.02 c	37.72 ± 3.16 e	36.22 ± 0.01 d	34.54 ± 1.23 b	32.79 ± 0.12 a
random	3.24 ± 0.26 e	0.25 ± 0.81 d	1.74 ± 1.12 cd	0.44 ± 0.00 a	1.16 ± 0.01 c	0.63 ± 0.35 ab	1.02 ± 0.18 c	1.21 ± 0.01 c
*β*-turn	9.11 ± 0.56 d	6.71 ± 0.24 a	7.68 ± 1.51 ab	8.54 ± 0.97 c	8.46 ± 0.11 c	8.66 ± 1.80 c	8.43 ± 0.97 c	7.23 ± 1.10 ab
*β*-turn/*α*-helix intermediate protein conformation	14.95 ± 1.00 b	21.28 ± 3.83 d	16.22 ± 0.46 c	15.66 ± 0.29 bc	14.76 ± 3.72 b	15.14 ± 0.56 bc	13.57 ± 4.09 a	13.52 ± 3.72 a
*α*-helix	26.36 ± 0.67 a	27.66 ± 3.94 b	28.50 ± 5.14 b	32.61 ± 1.32 c	31.19 ± 2.16 c	32.47 ± 2.60 c	35.86 ± 3.71 d	38.36 ± 2.19 e

^1^ Notation of samples as in Table 2. ^2^ Different letters within the same row indicates significant difference (*p* < 0.05)**.**

**Table 8 foods-12-03112-t008:** Evolution of the % contribution of the bread crumb secondary protein structures and the ratio of ordered to amorphous starch structures during storage.

Secondary Structure Assignment	Storage Time	CB ^1^	BLS	BFDS	BFDSM
	**Days**	**%Area**	**%Area**	**%Area**	**%Area**
Tyr ring	0	6.92 ± 0.12 aB ^2^	7.29 ± 0.08 bC	6.64 ± 0.09 aA	6.91 ± 0.03 bB
	4	6.92 ± 0.16 aA	6.90 ± 0.08 aA	6.87 ± 0.09 bA	6.79 ± 0.07 bA
	6	-	7.03 ± 0.09 aC	6.79 ± 0.06 abB	6.16 ± 0.03 aA
*β*-sheet	0	33.97 ± 0.09 aB	33.54 ± 0.14 aB	36.35 ± 0.34 aC	32.56 ± 0.40 aA
	4	34.88 ± 0.54 aB	42.10 ± 0.16 cD	36.88 ± 0.70 aC	33.14 ± 0.22 aA
	6	-	36.65 ± 0.05 bB	36.16 ± 0.15 aA	45.50 ± 0.2 0 bC
random	0	4.15 ± 0.09 aC	1.86 ± 0.08 bA	1.51 ± 0.12 aA	2.22 ± 0.11 bB
	4	4.19 ± 0.41 aC	0.63 ± 0.02 aA	2.03 ± 0.03 aB	1.45 ± 0.10 aB
	6	-	0.46 ± 0.01 aA	1.71 ± 0.59 aB	1.35 ± 0.22 aAB
*β*-turn	0	5.15 ± 0.03 aA	5.13 ± 0.04 aA	5.68 ± 0.08 aB	5.13 ± 0.0 4 bA
	4	4.26 ± 0.40 aB	5.69 ± 0.14 bC	5.20 ± 0.05 aC	3.22 ± 0.15 aA
	6	-	7.35 ± 0.03 cB	4.96 ± 0.66 aA	5.59 ± 0.03 cA
*β*-turn/*α*-helix intermediate protein conformation	0	16.68 ± 0.13 aAB	17.08 ± 0.10 cBC	16.27 ± 0.27 aA	17.43 ± 0.15 bC
	4	18.53 ± 0.25 bC	14.09 ± 0.18 aA	16.99 ± 0.13 aB	21.03 ± 0.18 cD
	6	-	15.36 ± 0.13 bB	17.77 ± 1.36 aC	12.87 ± 0.10 aA
*α*-helix	0	33.14 ± 0.18 bA	35.10 ± 0.20 cB	33.53 ± 0.35 aA	35.75 ± 0.28 cB
	4	31.21 ± 0.6 4aAB	30.59 ± 0.10 aA	32.02 ± 0.46 aB	34.36 ± 0.29 bC
	6	-	33.15 ± 0.04 bB	32.58 ± 1.21 aB	28.52 ± 0.46 aA
		**Ratio of Ordered to Amorphous Starch, R_1047/1022_**			
		CB	BLS	BFDS	BFDSM
R_1047/1022_	0	0.694 ± 0.001 aAB	0.682 ± 0.001 aA	0.689 ± 0.002 aA	0.695 ± 0.000 aB
	4 or 6 ^3^	0.726 ± 0.002 bB	0.718 ± 0.003 bA	0.717 ± 0.004 bA	0.747 ± 0.001 bC
Rate of R_1047/1022_ increase (day^−1^)		+0.008 C	+0.006 B	+0.004 A	+0.009 D

^1^ Notation of samples as in Table 2. ^2^ Different lowercase letters correspond to significant differences within the same column for each conformation type (*p <* 0.05); Different uppercase letters correspond to significant differences within the same row (*p <* 0.05). ^3^ 4 days for CB sample and 6 days for BLS, BFDS and BFDSM samples.

## Data Availability

Data is contained within the article or Appendix A.

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
