# Peer review of "Enhancement of Textural and Sensory Characteristics of Wheat Bread Using a Chickpea Sourdough Fermented with a Selected Autochthonous Microorganism"

_foods, 2023, doi:10.3390/foods12163112_

Round 1

Reviewer 1 Report

The authors selected a strain from traditional Greed sourdough to ferment chickpea flour as a standrd starter culture to improve the properties of bread. They found the liquid and freeze-dried chickpea sourdough improved the specific volume, texture and sensory properties of bread. 

Isolation of specific strains from traditional food and applicaiton of them in commercial are meaningful for the combination of traditional food and morden food industry. The study provided experimental data for the combination of  traditional Greek sourdough and bakery foods. 

I have one suggsetion about the methods and figures. The authors can add a shematic illustration of the methods (including at least 2.3, 2.4, and 2.6) to make the process more clear to readers. And please add some pictures of the bread on the schematic illustration.

Author Response

Dear reviewer, below you will find our response to your comment.

I have one suggestion about the methods and figures. The authors can add a shematic illustration of the methods (including at least 2.3, 2.4, and 2.6) to make the process more clear to readers. And please add some pictures of the bread on the schematic illustration.

Response: A schematic illustration was added to the text (Figure 1, lines 113, 177)

Reviewer 2 Report

The manuscript titled "Enhancement of textural and sensory characteristics of wheat bread using a chickpea sourdough fermented with a selected autochthonous microflora" deals within the scope of the Foods Journal, by investigating an interesting topic of research. The manuscript provides a detailed and comprehensive insight into the impact of chickpea sourdough fermented by C. perfringens CP8 as a starter culture in wheat bread production.

Minor remarks:

Line 244: In the research by Aremero et al., there is no "consistency" parameter described.

Line 251: The authors should describe in more detail the process of separating the crust and crumbs and measuring the moisture in them.

Line 255: According to the presented results, only the color of the crust was measured and this should be stated. If the authors also have the results of measuring the crumb color, it would be interesting to present those results as well.

Line 380: Was the temperature of ~96 °C established by measurement during the experiment? If not, the statement should be supported by a reference.

Line 404 (Figure S1): It is a little unclear whether the results from Figure S1 represent the initial viability of the C. perfringens since wheat flour was also added, which was normally added after 2 h of fermentation. It is necessary to clarify this in the Material and methods section and state it in the title of the Figure S1.

Line 532 (Table 4): For solids, it is better to use cm3/g rather than ml/g.

Author Response

Line 244: In the research by Aremero et al., there is no "consistency" parameter described.

Response:  This parameter was calculated according to Fustier et al. (2007) https://doi.org/10.1016/j.jcs.2006.10.011. However, to avoid any confusion, we removed this parameter throughout the text as it was not important and was barely discussed.

Line 251: The authors should describe in more detail the process of separating the crust and crumbs and measuring the moisture in them.

Response: This information was added to the text (lines 246-254)

Line 255: According to the presented results, only the color of the crust was measured and this should be stated. If the authors also have the results of measuring the crumb color, it would be interesting to present those results as well.

Response: We only included the crust color in the results, and this information was added to the text on line 256. The crumb color was similar among all the tested samples, therefore we chose not to include it in the results as we already had a large amount of data.

Line 380: Was the temperature of ~96 °C established by measurement during the experiment? If not, the statement should be supported by a reference.

Response: The temperature mentioned is established through measurements taken during the experiment.

Line 404 (Figure S1): It is a little unclear whether the results from Figure S1 represent the initial viability of the C. perfringens since wheat flour was also added, which was normally added after 2 h of fermentation. It is necessary to clarify this in the Material and methods section and state it in the title of the Figure S1.

Response: This information was added to the text (Figure S1; 959)

Line 532 (Table 4): For solids, it is better to use cm3/g rather than ml/g.

Yes, it is preferable to use cm3/g instead of ml/g when measuring solids. However, due to the irregular shape of bread, it is more practical to measure its volume in ml/g. Hence, the benchtop laser-based scanner we used for loaf specific volume measurements provided results in ml/g.

Reviewer 3 Report

Dear Author,

I would like to appreciate your efforts for a such interesting study.

The paper matches the aim and scope of the Journal. The introduction is well written. Methods are sound. Results are properly discussed.

Just one comment:

-    - Why was baker’s yeast used for the formulation of control bread instead of 100% wheat flour sourdough? In my opinion, the use of 100% wheat flour sourdough might help in pointing out the contribution of chickpea to improving bread textural and sensory characteristics.

Author Response

Dear reviewer, below you will find our response to your comment.

-    - Why was baker’s yeast used for the formulation of control bread instead of 100% wheat flour sourdough? In my opinion, the use of 100% wheat flour sourdough might help in pointing out the contribution of chickpea to improving bread textural and sensory characteristics.

Response: The main objective of this research was to prepare a dry chickpea sourdough using a starter culture composed of autochthonous microorganisms. The aim was to compare the textural and sensory attributes of a typical wheat bread versus chickpea sourdough bread formulations. To accomplish that, we did not use 100% wheat flour sourdough, as we wanted to explore other options. However, we may consider this idea for future research.

Reviewer 4 Report

Here are my comments for the authors:

The manuscript: “Enhancement of textural and sensory characteristics of wheat bread using a chickpea sourdough fermented with a selected autochthonous microflora”, is complex and very spoken in some of its parts and fluid in others and is very rich in often obvious information that weighs down the reading.

The authors probably had a mass of data which should have been better organised in some of its parts needs to be streamlined. In the results there are whole sentences that could be brought back into the discussion, this would make it easier to read.

With reference to the purpose of the work, which has the use of autochthonous microorganisms as its purpose, I believe that the exclusive use of only the CP8 strain of C. perfrigens is very limiting, in addition the choice of using it with short fermentations (2+1 hours) is not correct in accordance with traditional chickpea sourdough procedures.

The approach to sensory analysis is incorrect from a methodological point of view. The qualitative-quantitative profile is correct, as the authors write, in fact it requires a trained panel (it would be advisable to include a bibliographic reference), but the acceptability of the product commented in the results, and not described in the materials and  methods, is incorrect in  the approach.

Advice to indicate overall quality instead of acceptability.

For sensory analysis, acceptability is always the result of a hedonic score expressed by a panel of consumers, therefore not trained judges and the number varies according to the experimental plan (https://doi.org/10.1016/j.foodqual.2005.07.002).

Line-32 organoleptic, please use the term sensory, it is a term that the scientific community considers obsolete

1)    Did the judges who participated in the sensory evaluation sign an informed

consent ?

2)    On what basis did you decide to ferment the chickpea sourdough for only 3 hours?

3)    Lines 133-135. Strains producing the higher amount of gas and more appreciated for flavor in preparative fermentations of chickpea flour, as evaluated by members of a sensory panel familiar  with that type of bread, were finally employed as a starter culture for chickpea sourdough  production,( this entire concept is out of context).

4)    Lines 244-247. Did you cut the piece of bread just before the analysis? Can you find a reference for this test? I’m not sure that the method is correct since you separated the crust from the crumb. I’d have run the test on the whole loaf. Please explain your choice. Can you also explain how you cut a slice 0,5 mm thick of crust, with which instrument.

5)    Line 254- Color  parameters -Crust or crumb color?

6)    Lines 519-527 the results should  be reported before any other comment. Move this phrase elsewhere it can be usefull if applicable.

7)    Lines 530-  561-562 -With—correct  to

8)    Line 544- microflora- obsolete term----uses microorganism  Also correct the title

9)    Lines 557-559- the color of the crust can be greatly affected by baking condition. Why didn’t’you measure crumb color?

10)          Lines 598-599-write the sentence like this: CB sample exhibited the highest increase (12,5%) of crust moisture while the…..

11)          Lines 605-610  The whole concept should be eliminated to make the results easier to read

12)          Line 641 correlated

13)          Line 643-please add “N” and correlation coefficient with significance level

14)          Figure 4-you should fit the curves to an adequate model to discuss of “kinetics”

15)          Table 6- also here you should fit data to Avrami model to discuss “kinetics”, simple means are not suited. They  just describe a trend

The 3.13 section is really too long and somewhat out of the scope of the research

Author Response

Dear reviewer, below you will find our responses to your comments.

With reference to the purpose of the work, which has the use of autochthonous microorganisms as its purpose, I believe that the exclusive use of only the CP8 strain of C. perfrigens is very limiting, in addition the choice of using it with short fermentations (2+1 hours) is not correct in accordance with traditional chickpea sourdough procedures.

Response: After conducting preliminary experiments, we decided to utilize the CP8 strain for bread production due to its most desirable physicochemical and sensory characteristics compared to the other isolated strains. Additionally, the traditional chickpea sourdough is considered ready when it forms foam and doubles in volume, which typically takes approximately 18 hours of fermentation. Our objective was to expedite the fermentation process by employing an isolated strain from autochthonous microorganisms. In this regard, after only three hours of fermentation, the chickpea sourdough fermented by C. perfringens CP8 had formed foam and doubled in volume, indicating that it was ready to use.

The approach to sensory analysis is incorrect from a methodological point of view. The qualitative-quantitative profile is correct, as the authors write, in fact it requires a trained panel (it would be advisable to include a bibliographic reference), but the acceptability of the product commented in the results, and not described in the materials and methods, is incorrect in the approach. Advice to indicate overall quality instead of acceptability. For sensory analysis, acceptability is always the result of a hedonic score expressed by a panel of consumers, therefore not trained judges and the number varies according to the experimental plan (https://doi.org/10.1016/j.foodqual.2005.07.002).

Response: For the preference test, all samples were given simultaneously to the non-trained panelists, who were then asked to mark the number of the sample they preferred on the provided sheet. This procedure was adapted from Lawless & Heymann (2010) and additional information about the procedure used for the end product preference test was included in the text along with bibliographic references (lines 344-345)   

Line-32 organoleptic, please use the term sensory, it is a term that the scientific community considers obsolete

Response: The change has been made (line 31)

1)    Did the judges who participated in the sensory evaluation sign an informed consent?

Response: We provided full and transparent disclosure of the study requirements and potential risks. Participants were not coerced to participate. Additionally, all participants provided either written or verbal consent prior to sensory evaluation. We take privacy concerns very seriously, which is why we ensured that participant data would not be released without their knowledge and explicit consent. The tested end- products in this study were safe for consumption and the participants were given the option to withdraw from the study at any time without being asked for explanations of their decision.

2)    On what basis did you decide to ferment the chickpea sourdough for only 3 hours?

Response: Traditionally, chickpea sourdough is deemed ready when it forms foam and doubles in volume. Therefore, after 3 hours of fermentation, the chickpea sourdough fermented by the starter culture was considered ready for use. In the traditional method of making chickpea sourdough, autochthonous microorganisms exist in spores, resulting in a slow fermentation process that lasts approximately 18 hours. However, when chickpea sourdough is fermented using a starter culture, the microorganisms are activated, leading to a faster fermentation process that takes only 3 hours.

3)    Lines 133-135. Strains producing the higher amount of gas and more appreciated for flavor in preparative fermentations of chickpea flour, as evaluated by members of a sensory panel familiar with that type of bread, were finally employed as a starter culture for chickpea sourdough production (this entire concept is out of context).

Response: Lines have been removed from the text. 

4)    Lines 244-247. Did you cut the piece of bread just before the analysis? Can you find a reference for this test? I’m not sure that the method is correct since you separated the crust from the crumb. I’d have run the test on the whole loaf. Please explain your choice. Can you also explain how you cut a slice 0,5 mm thick of crust, with which instrument.

Response: The breads were sliced just before the analysis. To perform a texture profile analysis on the crumb and a puncture test on the crust, it was necessary to separate the crust from the crumb.

References for puncture test https://doi.org/10.1021/jf800522c; https://doi.org/10.1016/j.lwt.2018.02.065; https://doi.org/10.3390/foods10081832

Regarding the thickness of the crumb, there was an error in the text. The slice had a thickness of 5mm. This information has been corrected in the text (line 239).

5)    Line 254- Color  parameters -Crust or crumb color?

Response: “Color parameters of the crust”. (line 256)

6)    Lines 519-527 the results should  be reported before any other comment. Move this phrase elsewhere it can be usefull if applicable.

  Response: The given phrase has been relocated to lines 548-557

7)    Lines 530-  561-562 -With—correct  to

 Response: Corrections have been done (lines 540; 564; 565)

8)    Line 544- microflora- obsolete term----uses microorganism  Also correct the title

 Response: The microflora has been replaced (title, line 531) 

9)    Lines 557-559- the color of the crust can be greatly affected by baking condition. Why didn’t’you measure crumb color?

 Response: Crust color can be influenced by baking conditions, and therefore, all the tested bread fortifications baked under the same conditions. The crumb color appeared similar among the tested samples; thus, we did not include it since we already had a considerable amount of data.

10)          Lines 598-599-write the sentence like this: CB sample exhibited the highest increase (12,5%) of crust moisture while the…..

 Response: Chances have been made ( line 603)

11)          Lines 605-610  The whole concept should be eliminated to make the results easier to read

 Response: Those lines have been removed from the text

12)          Line 641 correlated

Response: Correction have been done (line 645)

13)          Line 643-please add “N” and correlation coefficient with significance level

  Response: This information was added to the text ( lines 647-648)

14)          Figure 4-you should fit the curves to an adequate model to discuss of “kinetics”

Response: The term “kinetics” has been removed in most cases throughout the text and when it was appropriate it was replaced by the word ‘evolution’ (line 634; 671).

15)          Table 6- also here you should fit data to Avrami model to discuss “kinetics”, simple means are not suited. They  just describe a trend

 Response: Correction have been made (line 671)

The 3.13 section is really too long and somewhat out of the scope of the research

Response: The section is now shorter. The importance of the FTIR measurements has been highlighted in the introduction section (lines 95-99)

Reviewer 5 Report

The manuscript presents an innovative approach that explores the potential of a traditional Greek sourdough, derived from chickpea flour fermentation through an indigenous culture, as an enhancer for wheat bread production. The research addresses a novel problem by investigating the utilization of this autochthonous culture to improve the quality and shelf life of wheat bread. The study’s significance is further highlighted by its multifaceted analysis, encompassing microbial identification, texture analysis, calorimetry, and sensory evaluation. The manuscript is well-written and effectively communicates the research findings. However, I would like to offer a few minor suggestions to further enhance its clarity and coherence.

Abstract

Provided abstract exceeds the prescribed word limit for author guidelines. Additionally, it lacks the inclusion of specific values or their changes pertaining to the primary findings. Abbreviations or acronyms should either be expanded or avoided in abstracts. The clarification of acronyms when first introduced should be validated throughout the entirety of the text. It is recommended that authors incorporate a statement concerning the safety of the employed strains, as potential doubts may arise and also put the strain name in italics.

Introduction

The introduction is well-written, but it seems to be too long. I also suggest moving the results of the preliminary research from the part discussing the research objectives to the section where the methods are described. This would provide a clearer justification for why we employed those methods.

Material and methods

The absence of a dedicated section detailing the materials employed for the study is notable. Furthermore, certain materials utilized, such as wheat flour and partially maltodextrin, have not been precisely enumerated. It is also advisable to amend the citation style in line 140 and verify the accurate usage of the Celsius degree symbol throughout the entire text. It is necessary to specify whether the color measurement pertained to the crust or the crumb of the breads. In the HPLC method description, I suggest first detailing the sample preparation procedure before outlining the protocol itself.

Results and discussion

The results are thoroughly described and well discussed. I recommend enhancing the readability of Figure 2 by dividing it into two or three parts (a, b, c). Furthermore, to condense the substantial volume of presented data, it might be advantageous to employ Principal Component Analysis (PCA).

Author Response

Dear reviewer, below you will find our responses to your comments.

Abstract

Provided abstract exceeds the prescribed word limit for author guidelines. Additionally, it lacks the inclusion of specific values or their changes pertaining to the primary findings. Abbreviations or acronyms should either be expanded or avoided in abstracts. The clarification of acronyms when first introduced should be validated throughout the entirety of the text. It is recommended that authors incorporate a statement concerning the safety of the employed strains, as potential doubts may arise and also put the strain name in italics.

Response: The abstract is 200 words which is the maximum that the journal requires. Specific values were not included to avoid exceeding the prescribed limit. Abbreviations were also avoided in the abstract. Additionally, a statement regarding the safety of the employed strains has been added for clarity (lines 14-15)

Introduction

The introduction is well-written, but it seems to be too long. I also suggest moving the results of the preliminary research from the part discussing the research objectives to the section where the methods are described. This would provide a clearer justification for why we employed those methods.

Response: This section has been shortened and the preliminary research findings have been excluded.

Material and methods

The absence of a dedicated section detailing the materials employed for the study is notable. Furthermore, certain materials utilized, such as wheat flour and partially maltodextrin, have not been precisely enumerated. It is also advisable to amend the citation style in line 140 and verify the accurate usage of the Celsius degree symbol throughout the entire text. It is necessary to specify whether the color measurement pertained to the crust or the crumb of the breads. In the HPLC method description, I suggest first detailing the sample preparation procedure before outlining the protocol itself.

Response: The "Raw materials" section has been included in the text (lines 101-104). Corrections related to the citation style in line 140 and the proper usage of the Celsius degree symbol have been made throughout the entire text. Additionally, information regarding the measurement of color specifically for the crust has been added. (line 256). Regarding the description of HPLC method the section "Sample preparation" has been moved prior to "Equipment and analytical procedure" section (lines 261-267)

Results and discussion

The results are thoroughly described and well discussed. I recommend enhancing the readability of Figure 2 by dividing it into two or three parts (a, b, c). Furthermore, to condense the substantial volume of presented data, it might be advantageous to employ Principal Component Analysis (PCA).

Response: Figure 2 is now renumbered as Figure 3, since a flow chart (Fig. 1) has been inserted in the revised text; Figure 3 has been now divided into two parts, denoted as (a) and (b) (lines 441-450).

Although we intended to conduct a Principal Component Analysis, the rather limited number of dough/bread formulations tested prevented us from doing so. In future research, we plan to conduct Principal Component Analysis to test and develop a framework for differentiation among samples.